# Contextual factors associated with contraceptive utilization and unmet need among sexually active unmarried women in Kenya: A multilevel regression analysis

**Bennett Nemser** [1]*, **Nicholas Addofoh**[2]

**1** University of the Western Cape, Cape Town, South Africa, **2** Episcopal Relief & Development, New York, NY, United States of America

* bnemser@gmail.com

**Data Availability Statement:** Performance Monitoring and Accountability (PMA) surveys. Data is available for download from their website. PMA Kenya Datasets • Request datasets: https://www.

## Abstract

### Background

Unmarried women who report less recent sexual intercourse (>30 days from survey enumeration) are largely excluded from global health monitoring and evaluation efforts. This study investigated level and contextual factors in modern contraceptive utilization and unmet need within this overlooked female subpopulation in Kenya from 2014 to 2019.

### Methods

This study analyzed data from the Performance Monitoring and Accountability (PMA) survey in Kenya, a nationally representative survey of female respondents, to understand the level and contextual factors for family planning utilization and unmet need within female subgroups including married, unmarried sexually active (defined as sexual intercourse within 30 days of survey enumeration), and unmarried with less recent sexual intercourse (defined as sexual intercourse 1–12 months prior to survey enumeration). The analysis included multilevel regression modeling to assess correlates on outcomes of modern contraceptive prevalence rate (mCPR), unmet need, and recent emergency contractive pill (ECP) use, which is a unique PMA question: "*Have you used emergency contraception at any time in the last 12 months?*".

### Results

Cumulatively, the surveys enumerated 19,161 women and this weighted analysis included 12,574 women aged 15–49 from three female subgroups: 9,860 married women (78.4%), 1,020 unmarried sexually active women (8.1%), and 1,694 unmarried women with less recent sexual intercourse (13.5%). In 2019, while controlling for covariates, unmarried women with less recent sexual intercourse exhibited statistically significant differences (p-value<0.02) in current mCPR, mCPR at last sexual intercourse, unmet need for modern contraceptives, and recent ECP use. As compared to an unmarried woman with less recent sexual intercourse (i.e., reported sex 1–12 months prior to survey), the odds of an unmarried sexually active woman (i.e., reported sex within last 30 days of survey) currently using

pmadata.org/data/request-access-datasets 2019
Phase 1 (2019) - Household & Female Survey:
Suggested citation: International Centre for
Reproductive Health Kenya (ICRHK); the Bill &
Melinda Gates Institute for Population and
Reproductive Health at the Johns Hopkins
Bloomberg School of Public Health; and Jhpiego.
Performance Monitoring for Action (PMA) Kenya
Phase 1: Household and Female Survey,
PMA2019/KE-P1-HQFQ. 2019. Kenya and
Baltimore, Maryland, USA. https://doi.org/10.
34976/4swk-g935 Phase 1 (2019) - Service
Delivery Point Survey: Suggested citation:
International Centre for Reproductive Health Kenya
(ICRHK); the Bill & Melinda Gates Institute for
Population and Reproductive Health at the Johns
Hopkins Bloomberg School of Public Health; and
Jhpiego. Performance Monitoring for Action (PMA)
Kenya Phase 1: Service Delivery Point Survey,
PMA2019/KE-P1-SQ. 2019. Kenya and Baltimore,
Maryland, USA. https://doi.org/10.34976/75jb-
n619 2017 Round 6 (2017) - Household & Female
Survey: Suggested citation: International Centre for
Reproductive Health Kenya (ICRHK); and the Bill &
Melinda Gates Institute for Population and
Reproductive Health at the Johns Hopkins
Bloomberg School of Public Health. Performance
Monitoring and Accountability 2020 (PMA2020)
Kenya Round 6: Household and Female Survey
(Version #), PMA2017/KE-R6-HQFQ. 2017. Kenya
and Baltimore, Maryland, USA. https://doi.org/10.
34976/mke4-va78 Round 6 (2017) - Service
Delivery Point Survey: Suggested citation:
International Centre for Reproductive Health Kenya
(ICRHK); and the Bill & Melinda Gates Institute for
Population and Reproductive Health at the Johns
Hopkins Bloomberg School of Public Health.
Performance Monitoring and Accountability 2020
(PMA2020) Kenya Round 6: Service Delivery Point
Survey (Version #), PMA2017/KE-R6-SQ. 2017.
Kenya and Baltimore, Maryland, USA. https://doi.
org/10.34976/6zm0-gj36 2014 Round 2 (2014) -
Household & Female Survey: Suggested citation:
International Centre for Reproductive Health Kenya
(ICRHK); and the Bill & Melinda Gates Institute for
Population and Reproductive Health at the Johns
Hopkins Bloomberg School of Public Health.
Performance Monitoring and Accountability 2020
(PMA2020) Kenya Round 2: Household and
Female Survey, PMA2014/KE-R2-HQFQ. 2014.
Kenya and Baltimore, Maryland, USA. https://doi.
org/10.34976/bryq-pf28 Round 2 (2014) - Service
Delivery Point Survey: Suggested citation:
International Centre for Reproductive Health Kenya
(ICRHK); and the Bill & Melinda Gates Institute for
Population and Reproductive Health at the Johns
Hopkins Bloomberg School of Public Health.

modern contraceptives was 2.28 (95% CI: 1.64, 3.18), using modern contraceptives at last sexual intercourse was 1.44 (95% CI: 1.06, 1.95), and having an unmet need for modern contraceptives was 2.01 (95% CI: 1.29, 3.13) while controlling for covariates. The odds of a married woman using ECP during the last 12 months was 0.60 (95% CI: 0.44, 0.82) as compared to an unmarried woman with less recent sexual intercourse. In 2019, unmarried women with less recent sexual intercourse reported the highest rate of ECP use during the last 12 months at 13.5%, which was similar for unmarried sexually active women at 13.3%. Since 2014, summary measures of unmet need and total demand for modern contraceptives increased for unmarried women with less recent sexual intercourse, but declined for the other female subgroups.

## Conclusion

In Kenya, unmarried women with less recent sexual intercourse exhibited significantly different contraceptive utilization, unmet need, and recent emergency contraceptive use. Moreover, changes over time in key family planning indicators were asymmetrical by female subgroup. This study identifies an important monitoring gap regarding unmarried women with less recent sexual intercourse. Evidence dissemination by the global measurement community for these unmarried women is exceedingly scarce; therefore, developing an inclusive research agenda and actionable information about these marginalized women is needed to enable targeted planning and equitable service delivery.

## Introduction

Access to high-quality family planning services has been identified as one of the most cost-effective strategies to improve health and development outcomes for women and their households [1]. Since the London Summit on Family Planning in 2012, multiple global efforts have sought to strengthen and provide equitable access to family planning services, including high-impact, low-cost methods, such as contraceptive implants and emergency contraceptives [2–5]. Over the last decade, modern contraceptive use has substantively increased in low- and middle-income countries (LMICs); however, the number of women with unmet need–where she wants to avoid pregnancy but is not using a modern contraceptive method—is also rising [6, 7]. Each year, 218 million women have an unmet need for modern contraceptives and approximately 111 million pregnancies are unintended [7]. Differences between married and unmarried women, who were sexually active in the last 30 days, are well documented with married women typically exhibiting lower contraceptive use and unmet need [8, 9]. However, family planning indicators for unmarried women with less recent sexual intercourse (>30 days) are largely unreported.

Women who want to avoid pregnancy, but are not using modern contraceptive methods, account for 77% of unintended pregnancies [7]. Emergency contraception, which is administered within a few days after sexual intercourse, can help prevent pregnancies due to non-use, failure or misuse of contraceptive, or situations of rape or coerced sex [10, 11]. While the copper intrauterine device is considered an emergency contraceptive method; the leading option is emergency contraceptive pills (ECP), which are oral contraceptive pills for women to use as soon as possible (up to 5 days) after sexual intercourse to prevent unwanted pregnancy [11]. ECP has a pregnancy prevention rate ranging from 56% to 95% if promptly used [12–16]. Suitably, ECP was selected as one of 13 high-impact, low-cost commodities by the UN

Performance Monitoring and Accountability 2020 (PMA2020) Kenya Round 2: Service Delivery Point Survey, PMA2014/KE-R2-SQ. 2014. Kenya and Baltimore, Maryland, USA. https://doi.org/10.34976/4kf2-t680.

**Funding:** The author(s) received no specific funding for this work.

**Competing interests:** The authors have declared that no competing interests exist.

Commission on Life-saving Commodities for Women and Children (UNCoLSC) [17]. ECP use is highest among two groups of women: aged 20–24 years and unmarried sexually active [18, 19]. ECP is safe for over-the-counter sale and often available from a pharmacist or drug seller without a prescription [20].

With a population of approximately 47 million, including 24 million women, Kenya is one of the most populous countries in sub-Saharan Africa and classified as a lower-middle income economy [21, 22]. Approximately 59.7% of women are currently married, which includes married or in union (i.e., living together) with a male partner [23]. Based on Kenya's most recent Demographic and Health Survey (DHS) in 2014 [23], the modern contraceptive prevalence rate (mCPR) was 39.1% for all women (including the subgroups of married women at 53.2% mCPR and unmarried sexually active women at 60.9% mCPR), which is one of the highest in SSA; however, ECP use was not reported. Correspondingly, unmet need for all women was 12.8% (including 17.5% for married women and 26.4% for unmarried sexually active women), which is relatively low for SSA. Kenya has implemented policies to reduce barriers to access family planning, such as policies enacted in 2013 to effectively eliminate family planning user fees as well as other public outpatient costs [24]. ECP is free at public health facilities and available for purchase without prescription in private pharmacies [25–27]. Since the last DHS was conducted over seven years ago, other data sources are needed to investigate recent changes in family planning practices in Kenya. This analysis utilized data from the Performance Monitoring and Accountability (PMA) survey [28], which was a nationally representative survey on family planning usage, knowledge, and experience of women. In addition, PMA incorporated a unique ECP question: "*Have you used emergency contraception at any time in the last 12 months*?". This question has a longer recall period than the traditional 'current use' ECP indicator, which underestimates the scale of ECP usage [11].

For family planning indicators, 'sexually active' is most commonly defined as a woman having sexual intercourse within one month (four weeks or 30 days) prior to the day of survey enumeration [29–32]. Sexual activity within one month is dramatically different between married and unmarried women. The proportion of married women who were sexually active (within one month) ranges from 50% to 91% across countries in SSA, while the sexual activity of unmarried women exhibits a lower range of 1% to 39% [30]. According to Kenya's DHS report in 2014, 79.5% of married women and 6.8% of unmarried were sexually active [30]. Research by Dasgupta et al. indicates when extending the definition of sexual activity beyond one month (e.g., 3 months or 12 months) the proportion of unmarried women who are considered sexually active increases drastically, while married women exhibit a modest increase [30]. As compared to married women, sexual encounters for unmarried women can be sporadic and unpredictable [31, 33]. Extending the time interval since last sexual activity for unmarried women can highlight the contraceptive needs of an underreported female subpopulation at risk of unintended pregnancies.

This study aims to evaluate the level in modern contraceptive utilization and unmet family planning needs among female subpopulations in Kenya: married or in union (i.e., living together); unmarried and sexually active within the past 30 days prior to survey (labeled as UA-30days); and unmarried and sexually active between 1–12 months prior to the survey (labeled as UA-12months). The latter, unmarried sexually active women with less recent sexual intercourse (between 1–12 months prior to survey), are underreported by the global health measurement community. Moreover, the analysis utilizes the unique survey design of current PMA questionnaires to investigate the level of recent emergency contraceptive use (within 12 months prior to survey). Lastly, the study assesses the relative effect of contextual factors (e.g., female subgroup, demographics, socioeconomic status) on these family planning outcomes and how these relationships changed over time in Kenya.

## Methods

### Study setting

PMA/Kenya was a nationally and county-level representative survey in Kenya from 2014 to 2019 that used a multi-stage stratified cluster design with urban-rural classification and geographic county as strata. The survey was enumerated in nine counties in 2014 and 11 counties in 2017 and 2019. Within each county, the sample of enumeration areas (EA) was selected by the Kenya National Bureau of Statistics using its master sample frame to provide a representative estimate of modern contraceptive prevalence rate (mCPR). Within each EA, 42 households were randomly selected for enumeration. Within each household, all eligible females aged 15–49 were designated for interview. Enumeration was conducted by local female residents using mobile technology for rapid data collection and quality control.

The sample of service delivery points (SDPs) include both public and private facilities where the catchment area falls within the EA boundary. Public facilities include health posts, primary health centers and the district hospitals. Up to three private facilities are randomly selected for enumeration if providing adequate maternal, reproductive, or general health services. Female resident enumerators survey private facilities, while enumeration supervisors survey public SDPs. For more details on PMA sampling and methodology see Zimmerman et al. [34].

### Data source and measurement

This study used data from three questionnaires: household, female, and service delivery point. The household questionnaire outlines the household roster and socioeconomic measures. The female questionnaire includes marital status, recency of sexual activity, family planning use, contraceptive knowledge, and recent experience with healthcare providers. The SDP questionnaire addresses the type of facility, service offerings, commodity availability (i.e., stock-outs), and fee structure among others.

In Kenya, PMA enumeration began in 2014 with annual or semi-annual cycles of data collection. For this study, data from 2014, 2017, and 2019 were analyzed. The 2017 survey was selected as the midpoint, because PMA introduced the question on emergency contraceptive use within the last 12 months in the 2017 questionnaire. Thus, 2017 will act as the baseline data point for the ECP model. In 2017 and 2019, PMA included an additional two counties, Kakamenga and West Pokot; however, this study restricted the analysis to women in the original nine counties to ensure comparability with 2014 data. Table 1 provides additional enumeration details for each PMA survey.

**Table 1. Description of PMA survey enumeration, PMA/Kenya (2014, 2017, 2019).**

|  | **2014** | **2017** | **2019** |
|---|---|---|---|
| Time of Collection | May—July 2014 | Nov—Dec 2017 | Nov—Dec 2019 |
| PMA Cycle (and citation) | Round 1 [35] | Round 6 [36] | Round 8 [37] [i.e. "Phase 1"] |
| Counties | 9 | 11 | 11 |
| Enumeration Areas (EA) | 120 | 151 | 308 |
| Households (% Response Rate) | 4,530 (93.2%) | 6,106 (97.8%) | 10,378 (98.1%) |
| Females (% Response Rate) | 3,807 (95.9%) | 5,876 (99.0%) | 9,478 (98.7%) |
| Facilities (% Response Rate) | 263 (nr) | 417 (97.2%) | 945 (94.6%) |

References for description of PMA survey enumeration [35–37]

Nr = Not reported

PMA/Kenya was managed by the Ministry of Health in partnership with International Centre for Reproductive Health Kenya (ICRHK), National Council for Population and Development, and Kenya National Bureau of Statistics. Johns Hopkins University (USA) and Jhpiego provided general direction and technical support. PMA was funded by the Bill & Melinda Gates Foundation.

## Study variables

The conceptual framework for analysis is outlined in Fig 1.

**Female subgroups (marital status, sexual activity).** Female respondents are categorized into three subgroups based on marital status and recency of sexual activity:

- Unmarried and sexually active between 1–12 months prior to the survey (UA-12months);

- Unmarried and sexually active within past 30 days prior to survey (UA-30days); or

- Married or in union.

**Primary outcome measures.** This study modeled four outcome measures at the level of the individual female respondent.

- **Current mCPR:** The first model analyzed modern contraceptive rate (mCPR), which is the proportion of women 15–49 years old who are using (or partner using) a modern method of contraception at the time of the survey (or 'current' mCPR). Long-acting modern methods include intra-uterine device (IUD), implant, and sterilization (male and female), while short-acting methods include injectable, pill, emergency contraception, male or female condoms, diaphragm, lactational amenorrhea method (LAM), and the standard days/cycle

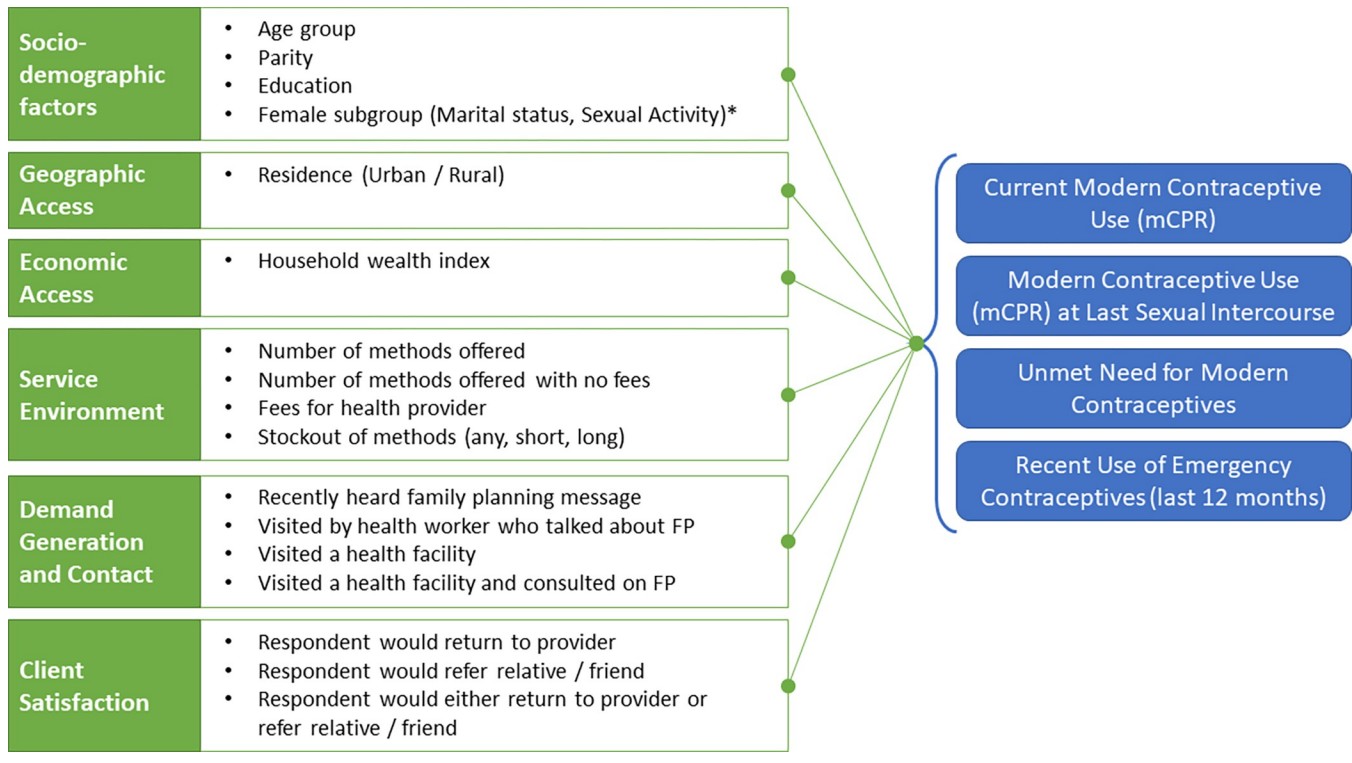

**Fig 1. Conceptual framework for analysis.**

beads method. Note: traditional methods, such as withdrawal and rhythm, are not considered modern methods.

- **mCPR at last sex:** The second model assessed mCPR at time of last sexual intercourse. According to Fabic et al. [31], using 'current' mCPR, which represents contraceptive use at time of survey, likely underreports mCPR for women with less recent sexual activity. A temporal misalignment occurs between 'current' use of modern contraceptives and last sexual intercourse, which could have been months prior. Measuring contraceptive use at last sexual activity can remedy this misalignment; however, historically few surveys include this type of question [31]. Data collection for this outcome indicator was only available for the 2019 PMA survey.

- **Unmet Need:** The third model explored unmet need for family planning defined as the proportion of fertile, sexually active women 15–49 years old who are not using contraception and do not want to become pregnant at any time (unmet need for limiting) or within the next two years (unmet need for spacing).

- **Recent Emergency Contraceptive use:** The final model analyzed emergency conceptive (ECP) use within the last 12 months, which is measured by the question (322a) "*Have you used emergency contraception at any time in the last 12 months*?" or if the female respondent is currently (question 302b) or recently (question 306b) using ECP.

  **Other outcome measures.**

- **Total Demand:** The proportion of female respondents or their partners who are currently using modern contraceptives (current mCPR) plus the female respondents with an unmet need for modern contraceptives (i.e., current mCPR plus unmet need).

- **Demand satisfied by modern contraceptives:** The proportion of women who are currently using modern contraceptives within the population of women demanding modern contraceptives (i.e., current mCPR divided by total demand).

## Other explanatory variables

See Table A.1 in S1 Appendix for definitions of additional explanatory variables evaluated including socio-demographic and wealth characteristics; healthcare delivery experience; and measures from the SDP questionnaire.

## Statistical analysis

First, univariate analyses were conducted for the explanatory variables across each survey period. Selected explanatory variables were assessed against the contraceptive use and unmet need outcome variables using bivariate regression analysis (not shown). To evaluate the relative effect of explanatory covariates (e.g., sociodemographic, female subgroup), multilevel regression models were built for each of the four outcome variables with EA, household, and female respondent as the respective levels to account for the hierarchical structure of the PMA/Kenya dataset (i.e., women nested within household and households nested within EAs). Relative to single-level regression, multilevel regression models will more accurately estimate standard errors of regression coefficients and properly assess statistical significance when analyzing hierarchical data.

To build the multilevel regression models, explanatory variables were added in a forward stepwise manner. Inclusion of explanatory variables was steered by the conceptual framework to showcase varying categories of influential factors. Variables were retained in the model if

they were statistically significant for an outcome variable (p-value < 0.05) or improve good-ness of fit (p-value > F value). To support interpretation of each covariate, the odds ratio (OR) was calculated and presented in Tables 4–7, which represents the odds of the outcome given exposure to the covariate. Statistical significance was assessed using 95% confidence intervals and p-values of 0.10, 0.05 and 0.01.

All analyses were performed in STATA software version 14 (College Station, TX, USA). Survey weights were applied to all estimates and sample sizes to adjust for each woman's likeli-hood of selection.

## Results

Cumulatively, the weighted analysis included 12,574 women aged 15–49 within the three female subgroups for the three enumeration periods–ranging from 2,956 women in 2014 to 5,957 women in 2019 (Table 2). About two-thirds of the female respondents were 15–34 years old, more than half had parity between 1–4 children, and approximately half completed only primary school or received no education. In 2019, approximately 62% of women lived in rural communities, which is fairly consistent across time periods. Most respondents were married, but a total of 1,694 respondents (13.5%) across the three enumeration periods were unmarried and sexually active between 1–12 months prior to the survey (UA-12months).

### Levels of contraceptive use by female subgroup

From 2014 to 2019, current mCPR increased for each female subgroup with the highest rate of change at 16 percentage points among unmarried women, who were sexually active within

**Table 2. Background characteristics for women aged 15–49, PMA/Kenya (2014, 2017, 2019).**

| Characteristics | Categories | 2014 | | 2017 | | 2019 | |
|---|---|---|---|---|---|---|---|
| | | n | % | n | % | n | % |
| **Age Group** | 15–24 | 792 | 26.8 | 1,028 | 28.1 | 1,588 | 26.7 |
| | 25–34 | 1,246 | 42.2 | 1,473 | 40.2 | 2,341 | 39.3 |
| | 35–44 | 710 | 24.0 | 903 | 24.7 | 1,535 | 25.8 |
| | 45–49 | 208 | 7.0 | 257 | 7.0 | 493 | 8.3 |
| **Parity** | None | 285 | 9.6 | 619 | 16.9 | 781 | 13.1 |
| | 1–2 | 1,210 | 40.9 | 1,401 | 38.3 | 2,272 | 38.1 |
| | 3–4 | 860 | 29.1 | 1,007 | 27.5 | 1,743 | 29.3 |
| | 5 plus | 601 | 20.3 | 634 | 17.3 | 1,161 | 19.5 |
| **Education** | Never | 114 | 3.9 | 109 | 3.0 | 198 | 3.3 |
| | Primary | 1,637 | 55.4 | 1,779 | 48.6 | 2,859 | 48.0 |
| | Secondary or more | 1,205 | 40.8 | 1,773 | 48.4 | 2,900 | 48.7 |
| **Household wealth quintile** | Lowest | 641 | 21.8 | 525 | 14.3 | 909 | 15.3 |
| | Middle lowest | 758 | 25.7 | 768 | 21.0 | 1,225 | 20.6 |
| | Middle | 548 | 18.6 | 761 | 20.8 | 1,378 | 23.1 |
| | Middle highest | 576 | 19.6 | 781 | 21.3 | 1,226 | 20.6 |
| | Highest | 423 | 14.4 | 826 | 22.6 | 1,219 | 20.5 |
| **Residence** | Urban | 1,145 | 38.7 | 1,390 | 38.0 | 2,266 | 38.0 |
| | Rural | 1,811 | 61.3 | 2,271 | 62.0 | 3,691 | 62.0 |
| **Female subgroups** | Unmarried sexually active (1–12 months) | 291 | 9.8 | 558 | 15.2 | 845 | 14.2 |
| | Unmarried sexually active (0–30 days) | 154 | 5.2 | 338 | 9.2 | 528 | 8.9 |
| | Married or in union | 2,511 | 84.9 | 2,765 | 75.5 | 4,584 | 77.0 |
| **Total Women** | | **2,956** | | **3,661** | | **5,957** | |

0–30 days of the survey (UA-30days) (Fig 2). In 2014, the highest current mCPR was among married women, but UA-30days eclipsed married women by 2019 with an mCPR of 62%. In terms of contraceptive method mix, married women heavily favored implants and injectables, while unmarried women preferred male condoms, implants, and injectables (Table 3). From 2014 to 2019, all female subgroups decreased short-acting modern methods and increased both long-acting modern methods (predominantly contraceptive implants) and traditional techniques (e.g., rhythm, withdrawal). In 2019, mCPR at last sexual intercourse for UA-12months was 12 percentage points higher than their current use of modern contraceptives (Fig 3), while UA-30days exhibited a three percentage point relative increase.

**Levels of unmet need and total demand by female subgroup.**   Unmet need for family planning indicated a complementary decline for married and UA-30days over time (Fig 4). Since 2014, unmet need for married and UA-30days decreased by 10 and 24 percentage points, respectively. However, UA-12months more than doubled from 6% to more than 13%. These changes over time were consistent in both unmet need for spacing and limiting (Table A.2 in S1 Appendix). Total demand (mCPR plus unmet need) for UA-30days was highest amongst the female subgroups at 85% and displayed a similarly proportioned decline over time for married women (Fig 5); however, UA-12months increased 16 percentage points to 57%. By 2019, the percent of total demand satisfied by modern contraceptives (Fig 6) was clustered between 73% and 80% for all female subgroups. UA-30days improved dramatically over time, but UA-12months declined since 2014.

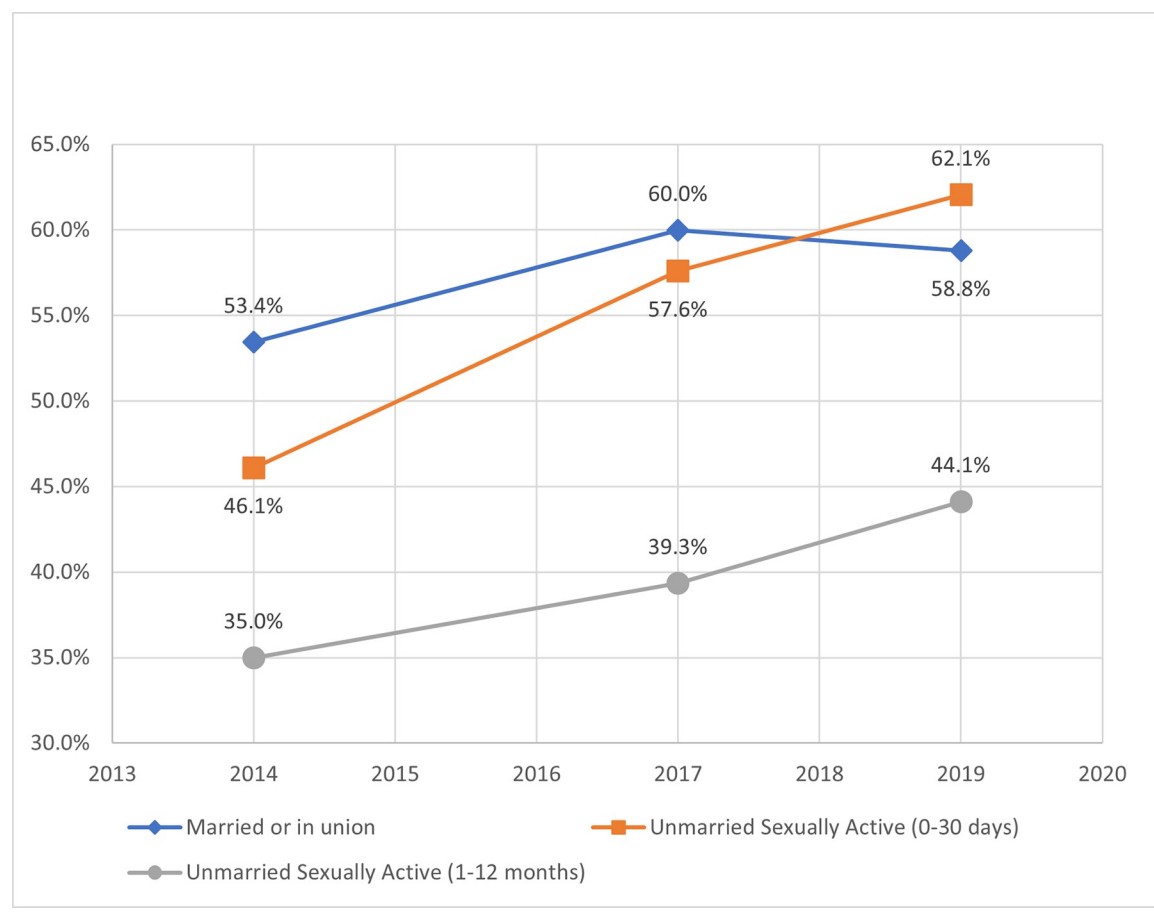

**Fig 2. Current modern contraceptive prevalence rate by female subgroup.**

**Table 3. Contraceptive method mix by female subgroup, PMA/Kenya (2014, 2017, 2019).**

| Female Population: | Unmarried Sexually Active (1–12 months) | | | | Unmarried Sexually Active (0–30 days) | | | | Married or In Union | | | |
|---|---|---|---|---|---|---|---|---|---|---|---|---|
| PMA2020 Survey Year: | 2014 | 2017 | 2019 | Change | 2014 | 2017 | 2019 | Change | 2014 | 2017 | 2019 | Change |
| Number of women using any contraceptives | 97 | 232 | 402 | | 69 | 201 | 354 | | 1,350 | 1,694 | 2,893 | |
| **Current Method Mix** | | | | | | | | | | | | |
| Female Sterilization | 3% | 2% | 2% | 0% | 0% | 0% | 2% | 2% | 4% | 4% | 5% | 1% |
| Male Sterilization | 0% | 0% | 0% | 0% | 0% | 0% | 0% | 0% | 0% | 0% | 0% | 0% |
| Implant | 17% | 20% | 24% | 8% | 16% | 20% | 28% | 11% | 18% | 34% | 38% | 19% |
| IUD | 5% | 1% | 2% | −3% | 2% | 1% | 4% | 2% | 5% | 3% | 5% | 0% |
| Injectable: intramuscular (IM) | 47% | 30% | 22% | −25% | 44% | 31% | 25% | −19% | 58% | 45% | 33% | −25% |
| Injectable: subcutaneous (SC) | 0% | 0% | 2% | 2% | 0% | 0% | 3% | 3% | 0% | 0% | 3% | 3% |
| Pill | 6% | 4% | 5% | −1% | 11% | 8% | 9% | −3% | 10% | 8% | 8% | −2% |
| Emergency Contraceptive Pills (ECP) | 9% | 13% | 10% | 1% | 9% | 14% | 4% | −5% | 1% | 1% | 1% | 0% |
| Male Condom | 14% | 23% | 22% | 9% | 17% | 21% | 19% | 2% | 3% | 2% | 3% | 0% |
| Female Condom | 1% | 0% | 0% | −1% | 0% | 1% | 0% | 0% | 0% | 0% | 0% | 0% |
| Diaphragm | 0% | 0% | 0% | 0% | 0% | 0% | 0% | 0% | 0% | 0% | 0% | 0% |
| Cycle beads | 0% | 0% | 1% | 1% | 0% | 1% | 1% | 1% | 0% | 1% | 1% | 0% |
| LAM[1] | 0% | 0% | 0% | 0% | 0% | 0% | 0% | 0% | 0% | 0% | 0% | 0% |
| Rhythm | 0% | 4% | 5% | 5% | 1% | 2% | 4% | 3% | 0% | 2% | 2% | 2% |
| Withdrawal | 0% | 2% | 2% | 2% | 0% | 1% | 1% | 1% | 0% | 0% | 1% | 1% |
| Other Traditional[2] | 0% | 0% | 2% | 2% | 0% | 0% | 2% | 2% | 0% | 0% | 2% | 2% |
| **Total** | **100%** | **100%** | **100%** | | **100%** | **100%** | **100%** | | **100%** | **100%** | **100%** | |
| **Injectable subtotal** | | | | | | | | | | | | |
| Injectable total (IM + SC)[3] | 47% | 30% | 24% | −23% | 44% | 31% | 28% | −16% | 58% | 45% | 35% | −22% |
| **Type of Method** | | | | | | | | | | | | |
| Long-acting modern method[4] | 24% | 23% | 29% | 5% | 18% | 22% | 33% | 16% | 27% | 41% | 47% | 20% |
| Short-acting modern method[5] | 76% | 71% | 62% | −14% | 81% | 75% | 60% | −21% | 73% | 56% | 48% | −25% |
| Traditional method[6] | 0% | 6% | 9% | 9% | 1% | 3% | 6% | 6% | 0% | 3% | 5% | 5% |

1. LAM: Lactational amenorrhea method.

2. Other traditional: all respondent-mentioned other methods.

3. IM: Intramuscular injectable; SC: subcutaneous injectable

4. Long-acting modern methods include intra-uterine device (IUD), implant, and sterilization (male and female)., while short-acting methods.

5. Short-acting modern methods include injectable, pill, emergency contraception, male or female condoms, diaphragm, lactational amenorrhea method (LAM), and the standard days/cycle beads method.

6. Traditional methods include rhythm method, withdrawal, and all respondent-mentioned other methods (labeled 'other traditional'):.

## Levels of recent emergency contraceptive use by female subgroup

In 2019, recent use of emergency contraceptive was highest for UA-12months at 13.5% along with the largest gain of 2.9 percentage points since 2017 (Fig 7). Married and UA-12months increased over time, while UA-30days exhibited a slight decline.

## Contextual factors influencing modern contraceptive use

Table 4 reports odds ratios of current mCPR while controlling for demographic, socioeconomic, and service delivery factors. In 2019, women who were younger, had more children, had higher levels of education, and lived in urban areas exhibited significantly higher rates of current modern contraceptive use. Since 2014, these factors affecting current mCPR typically became more pronounced, while the influence of household wealth was minimized. While

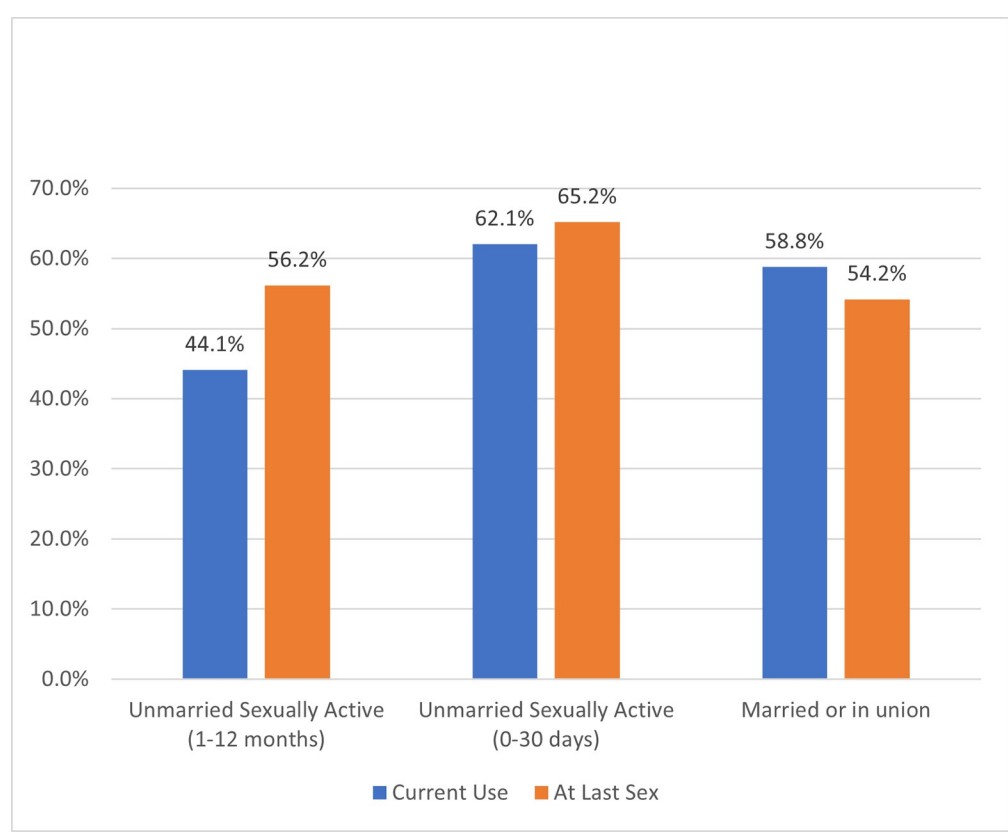

**Fig 3. Modern contraceptive prevalence rate at last sexual intercourse compared to current use mCPR by female subgroup (2019 survey).**

controlling for all other factors, UA-12months reported significantly lower rates of current modern contraceptive use relative to UA-30days. This discrepancy between unmarried groups grew more evident since 2014. Married women had significantly higher current mCPR relative to UA-12months in 2014, but that difference waned over time.

Hearing family planning messages in the community was a modest factor in 2014, but that influence dissipated by 2019. While not statistically significant, having at least one facility not charging fees in the enumeration area increased mCPR and improved the model's overall performance, so the indicator was retained across models. No other service delivery, patient contact or experience indicators were a significant factor in the regression models.

For mCPR at last sexual intercourse in 2019 (Table 5), there was a significant difference for female subgroups. UA-12months reported significantly higher modern contraceptive use at last sexual intercourse as compared to married women, but significantly lower than UA-30days. Relative to the current mCPR model (Table 4), the magnitude of effect (odds ratio) was smaller for last sexual intercourse when comparing contraceptive use between UA-30days and UA-12months. The other socio-demographic factors (e.g., age, parity, education, residence) exhibited similar influence on mCPR at last sex as compared to current mCPR; however, the odds ratio for parity was almost half the size.

## Contextual factors influencing unmet need

In 2019, unmet need for contraception was lower for women who were older, had received more education, were in higher income categories, and lived in urban areas (Table 6). Since

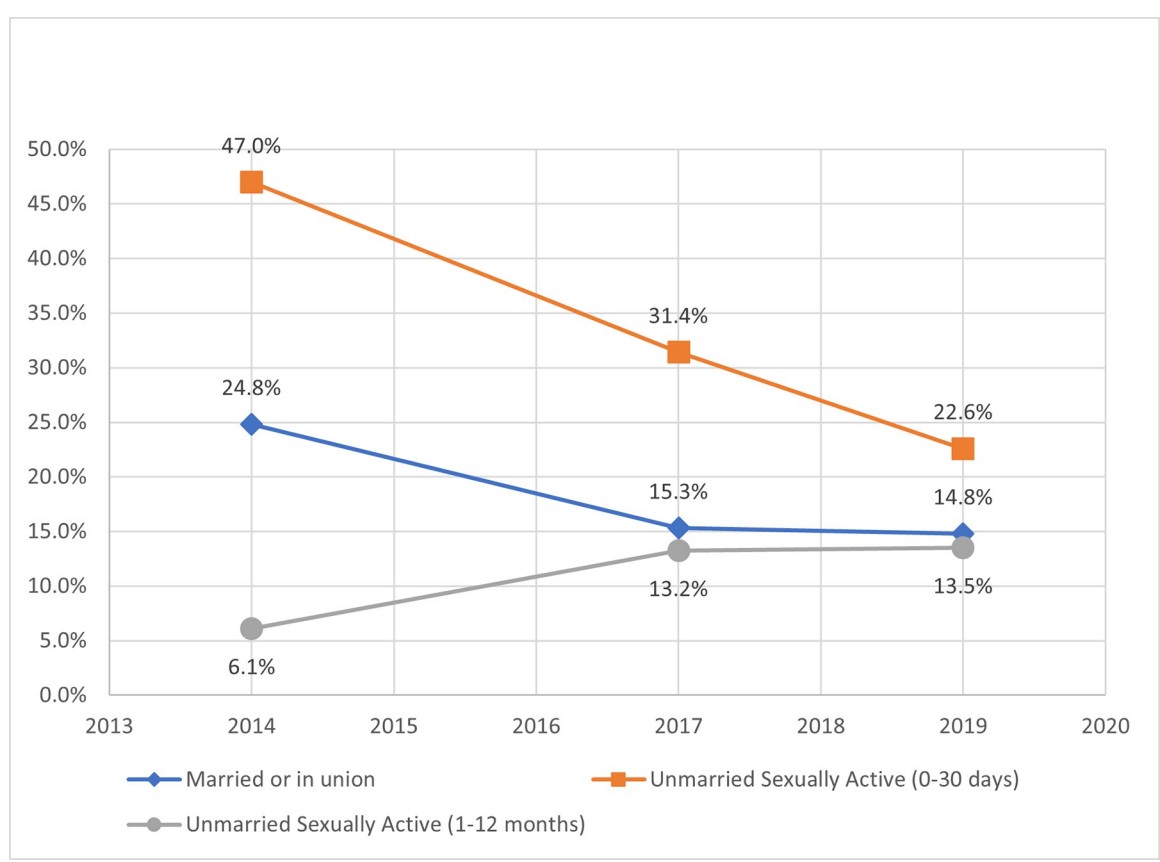

**Fig 4. Unmet need for modern contraceptives by female subgroup.**

2014, the age factor grew more prominent, while the education and economic influence weakened. When controlling for all other factors, unmet need for contraception was significantly different between female subgroups with UA-30days at approximately twice the odds of having unmet need relative to UA-12months in 2019. In 2014, married women had significantly higher unmet need compared to UA-12months, but that differential dissolved by 2014. Over time, the odds ratios relative to UA-12months have decreased, which indicates a narrowing gap between the unmet needs of UA-12months and the other two female subgroups.

### Contextual factors influencing recent emergency contraceptive use

For use of emergency contraceptives within the past 12 months, women aged 45–49 years or living in rural communities were less likely to use, while women in the two highest wealth quintiles were more likely to use ECP recently (Table 7). Since 2017, when this ECP question was first asked, the influence of parity, education and age (between 25–34 years) dissipated. For the female subgroups, married women had significantly lower recent usage of ECP relative to UA-12months in 2019, but not 2017. There was no difference in recent ECP usage between the two unmarried groups.

### Discussion

In Kenya, contraceptive dynamics differ by subpopulations of women and the change over time is asymmetric. While controlling for covariates, female subgroups—based on marital

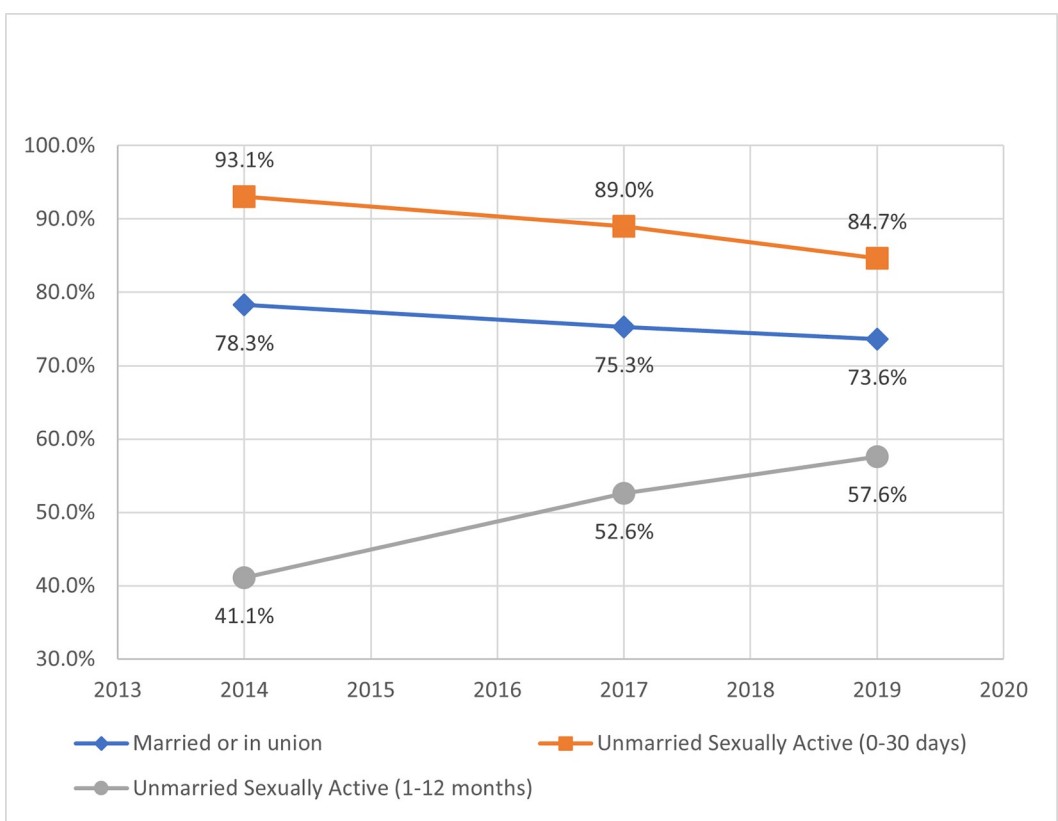

**Fig 5. Total demand for modern contraceptives by female subgroup.**

status and recency of sexual intercourse—exhibited significant differences for current mCPR, mCPR at last sexual intercourse, unmet need, and recent ECP use during the last 12 months. Since 2014, current mCPR has increased for all female groups, while unmet need, total demand, demand satisfied by modern contraceptives, and recent ECP use are moving in different directions depending on the female subgroup. Unmarried women who were sexually active between 1–12 months prior to survey (UA-12months) notably reported an increase in both unmet need and current mCPR as well as the highest rate of emergency contraceptive use in the last 12 months. Understanding the uniqueness of each female subpopulation will be critical for designing and implementing effective public health programs for all women.

Consistent with global trends [6, 7, 38], modern conceptive use grew across each female sub-population since 2014 with long-acting modern methods chosen over short-acting methods (Table 3). Short-term injectables were swapped for implants—particularly among married women—which matches a broader shift in contraceptive preferences [38]. Female condom use was minimal across all groups despite a comprehensive global push during the time period and the unique disease prevention and empowerment features of female condoms [4, 39]. In terms of accessibility, stock-out at health facilities was not a factor in mCPR, which is consistent with a similar analysis in Kenya [34]. However, women in rural communities had significantly lower mCPR, which may indicate geographic accessibility constraints [40]. While the timing differences between 'current' contraceptive use questions and the sexual activity interval of 12 months for unmarried women can generate relatively low CPR and high unmet need [31], mCPR at last sexual intercourse does not include this timing misalignment and this model exhibited similar influence by socio-demographic and economic factors relative to

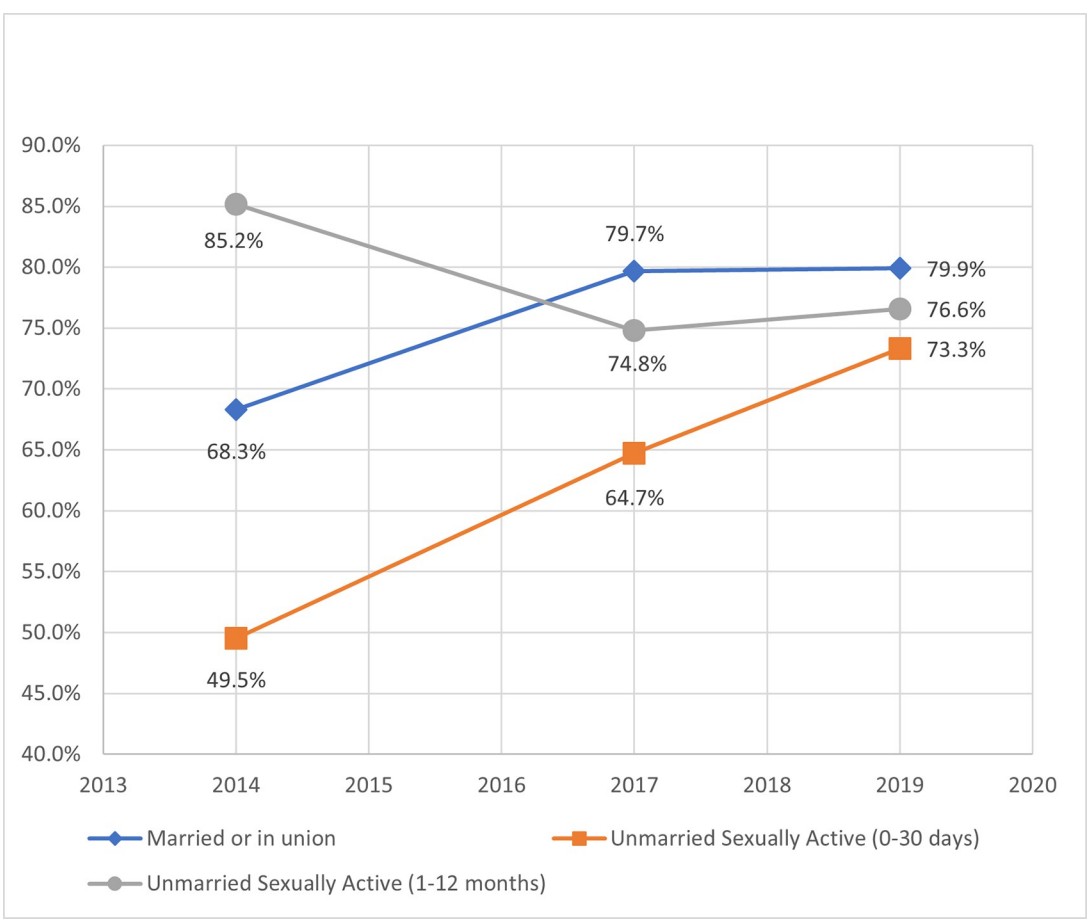

**Fig 6. Proportion of demand satisfied for modern contraceptives by female subgroup.**

current mCPR models in 2019. However, mCPR at last sexual intercourse for UA-12months was significantly higher than married women and significantly lower than UA-30days, which provides further evidence to consider these unmarried women as a distinct subpopulation for family planning analysis and program design. Developing monitoring systems to track family planning needs and trends in these unmarried women could be the mechanism for understanding how to build and adapt effective universal programming for women.

The female subpopulations diverged on unmet need for contraception. While unmet need doubled for unmarried women with less recent sex (UA-12months), the other female subgroups exhibited a sharp decline over the five-year period. For UA-12months, large increases in both mCPR and unmet need indicate a deep, previously unaddressed demand [8] for contraceptives and represents a critical area for further research and public health programming. The increase over time in unmet need may indicate a growing risk of unintended pregnancy for these unmarried women; however, the increased use of ECP allows for greater agency in response to unexpected sexual encounters. When controlling for female subgroup, the impact of education and wealth weakened since 2014, which may indicate positive effects of Kenya's 2013 policies to eliminate user fees on family planning services and public outpatient primary care [24]. Unmet need increased for rural and younger women, which are common underserved demographic groups across sub-Saharan Africa [9, 41, 42], and represent a practical target for improvement in family planning programs.

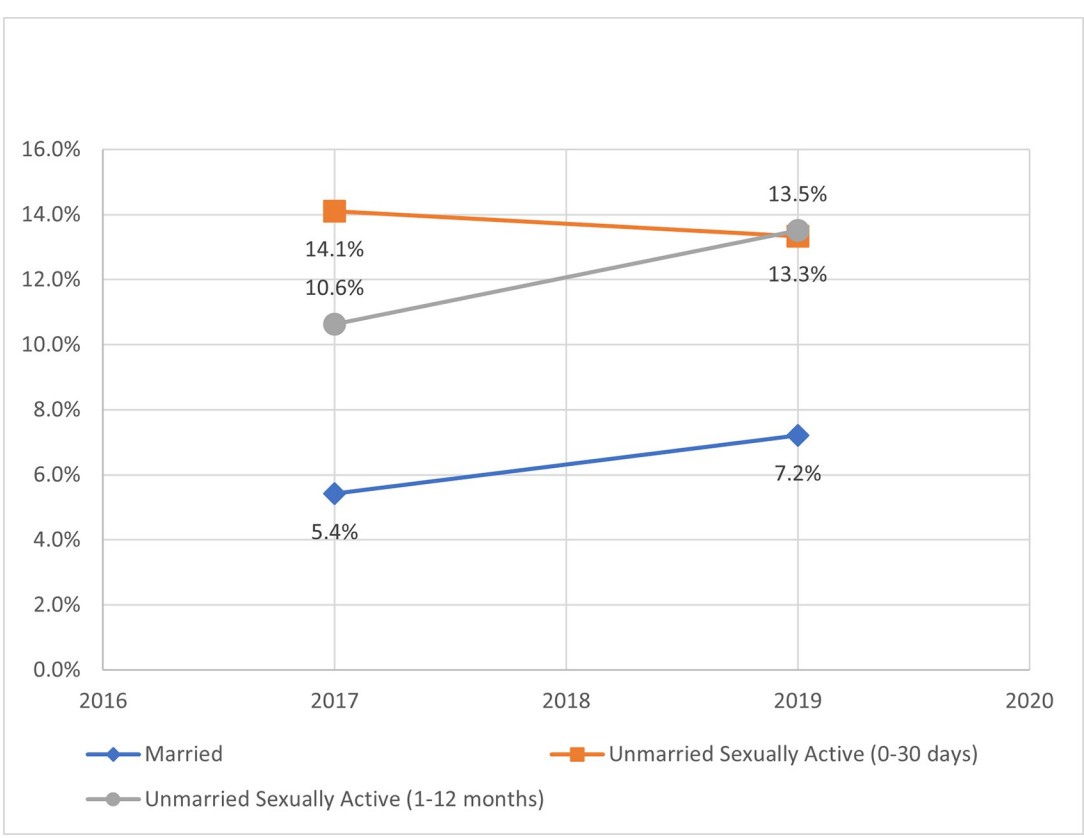

**Fig 7. Recent emergency contraceptive use (within the last 12 months) by female subgroup.**

When contraceptive needs are unmet, emergency contraceptives are an important option for women [9–11]. Consistent with other analyses [11], unmarried sexually active women in Kenya have significantly higher rates of recent ECP use (within the last 12 months) relative to married women (Table 7). However, this analysis also indicates a growth in ECP use over time–in particular for unmarried women with less recent sex (UA-12months), where more than one out of every eight women have used ECP in the last 12 months (Fig 7). In this study, unmarried sexually active women shifted away from shorter term modern methods, while increasing use of traditional methods, such as withdrawal, along with ECP. This matches other international reports of unmarried women increasing use of traditional methods [30] and utilizing ECP as back-up for possible method failure [33]. Sexual encounters for unmarried women can be sporadic or unpredictable [31, 33] and restricting recency measurements of sex to 30 days typically biases mCPR upward and unmet need downward [31]. In Kenya, this underreported subgroup of unmarried women with less recent sex exhibited similarly lower mCPR (current and at last sexual intercourse) and higher unmet need relative to their female counterparts–and is increasingly leveraging ECP access to prevent unwanted pregnancies. With ECP free in public health facilities and available without prescription at private pharmacies, approaches to further expand ECP uptake, such as targeted demand generation campaigns, may benefit this female subgroup in managing unintended pregnancies [27, 43].

To reach universal health coverage, public health policy makers and practitioners must understand the behavior and healthcare preferences of varying subpopulations in order to design and manage equitable healthcare delivery systems [5]. This study illustrates how

**Table 4. Logistic Regression model for Current Modern CPR by Survey Round, PMA/Kenya (2014, 2019).**

| Characteristic | Categories | KENYA 2014 (Baseline) | | | | | KENYA 2019 (Endline) | | | | |
| --- | --- | --- | --- | --- | --- | --- | --- | --- | --- | --- | --- |
| | | OR (95% CI) | | | P-value | | OR (95% CI) | | | P-value | |
| | | Odds Ratio | Lower bound | Upper bound | | | Odds Ratio | Lower bound | Upper bound | | |
| **Age Group** | 15–24 (ref) | 1 | | | | | 1 | | | | |
| | 25–34 | 1.11 | 0.87 | 1.43 | 0.387 | | 0.72 | 0.60 | 0.86 | 0.000 | *** |
| | 35–44 | 0.86 | 0.60 | 1.23 | 0.396 | | 0.47 | 0.38 | 0.59 | 0.000 | *** |
| | 45–49 | 0.27 | 0.17 | 0.42 | 0.000 | *** | 0.23 | 0.17 | 0.31 | 0.000 | *** |
| **Parity** | None (ref) | 1 | | | | | 1 | | | | |
| | 1–2 | 4.30 | 2.67 | 6.91 | 0.000 | *** | 4.65 | 3.67 | 5.90 | 0.000 | *** |
| | 3–4 | 6.95 | 4.12 | 11.70 | 0.000 | *** | 7.93 | 5.78 | 10.89 | 0.000 | *** |
| | 5 plus | 6.37 | 3.72 | 10.92 | 0.003 | *** | 8.80 | 6.15 | 12.59 | 0.000 | *** |
| **Education** | Never (ref) | 1 | | | | | 1 | | | | |
| | Primary | 3.17 | 2.01 | 4.99 | 0.000 | *** | 2.31 | 1.58 | 3.39 | 0.000 | *** |
| | Secondary or more | 3.86 | 2.50 | 5.97 | 0.000 | *** | 2.42 | 1.63 | 3.61 | 0.002 | *** |
| **Household wealth quintile** | Lowest (ref) | 1 | | | | | 1 | | | | |
| | Middle lowest | 1.29 | 0.99 | 1.68 | 0.064 | * | 0.95 | 0.76 | 1.18 | 0.626 | |
| | Middle | 1.39 | 1.05 | 1.85 | 0.024 | ** | 1.01 | 0.82 | 1.24 | 0.923 | |
| | Middle highest | 1.43 | 1.00 | 2.03 | 0.050 | * | 1.25 | 0.99 | 1.57 | 0.060 | * |
| | Highest | 1.48 | 0.92 | 2.36 | 0.104 | | 1.01 | 0.75 | 1.37 | 0.925 | |
| **Residence** | Urban (ref) | 1 | | | | | 1 | | | | |
| | Rural | 0.83 | 0.62 | 1.10 | 0.192 | | 0.76 | 0.59 | 0.96 | 0.025 | ** |
| **Female subgroup** | Unmarried sexually active (1–12 months) | 1 | | | | | 1 | | | | |
| | Unmarried sexually active (0–30 days) | 2.18 | 1.24 | 3.82 | 0.007 | *** | 2.28 | 1.64 | 3.18 | 0.000 | *** |
| | Married or in union | 1.63 | 1.08 | 2.48 | 0.021 | ** | 1.11 | 0.87 | 1.41 | 0.396 | |
| **Fees for health provider** | No (ref) | 1 | | | | | 1 | | | | |
| | Yes | 1.27 | 0.95 | 1.70 | 0.107 | | 0.90 | 0.68 | 1.19 | 0.452 | |
| **Recently heard FP message** | No (ref) | 1 | | | | | 1 | | | | |
| | Yes | 1.39 | 0.94 | 2.06 | 0.099 | * | 1.11 | 0.91 | 1.36 | 0.287 | |

P-value

\* P < 0.10

\*\* P < 0.05

\*\*\* P <0.01

Ref: Reference Category

unmarried women with slightly less recent sexual intercourse (0–30 days versus 1–12 months prior to survey) have significantly different contraceptive use and unmet needs for contraceptives. However, this subpopulation of unmarried women with less recent sex (UA-12months)–which constitutes 13% of women in this study–are routinely absent from data analysis and reporting. Effective data-informed decision-making requires iterative cycles of data collection, analysis, dissemination, and corrective action to improve service delivery [44], but the global health measurement community stops at data collection for unmarried women with less recent sex. Without timely information, decisions by public health policy makers and program managers are blinded to the needs of these unmarried women, which effectively marginalizes them through data processing. Moreover, this 'data marginalization' further exacerbates existing

**Table 5. Logistic regression model for mCPR at last sexual intercourse, PMA/Kenya 2019.**

| Characteristic | Categories | KENYA 2019 (Endline) | | | | |
|---|---|---|---|---|---|---|
| | | OR (95% CI) | | | P-value | |
| | | Odds Ratio | Lower bound | Upper bound | | |
| Age Group | 15–24 (ref) | 1 | | | | |
| | 25–34 | 0.74 | 0.62 | 0.88 | 0.001 | *** |
| | 35–44 | 0.55 | 0.44 | 0.68 | 0.000 | *** |
| | 45–49 | 0.35 | 0.26 | 0.47 | 0.000 | *** |
| Parity | None (ref) | 1 | | | | |
| | 1–2 | 2.54 | 2.04 | 3.17 | 0.000 | *** |
| | 3–4 | 4.27 | 3.15 | 5.77 | 0.000 | *** |
| | 5 plus | 4.61 | 3.33 | 6.40 | 0.000 | *** |
| Education | Never (ref) | 1 | | | | |
| | Primary | 1.88 | 1.23 | 2.90 | 0.004 | *** |
| | Secondary or more | 2.25 | 1.44 | 3.49 | 0.000 | *** |
| Household wealth quintile | Lowest (ref) | 1 | | | | |
| | Middle lowest | 1.02 | 0.81 | 1.27 | 0.882 | |
| | Middle | 1.22 | 0.96 | 1.54 | 0.101 | |
| | Middle highest | 1.43 | 1.08 | 1.88 | 0.012 | ** |
| | Highest | 1.44 | 1.03 | 2.04 | 0.035 | ** |
| Residence | Urban (ref) | 1 | | | | |
| | Rural | 0.68 | 0.50 | 0.91 | 0.009 | *** |
| Female subgroup | Unmarried sexually active (1–12 months) | 1 | | | | |
| | Unmarried sexually active (0–30 days) | 1.44 | 1.06 | 1.95 | 0.019 | ** |
| | Married or in union | 0.67 | 0.54 | 0.83 | 0.000 | *** |
| Fees for health provider | No (ref) | 1 | | | | |
| | Yes | 1.27 | 0.90 | 1.80 | 0.178 | |
| Recently heard FP message | No (ref) | 1 | | | | |
| | Yes | 1.13 | 0.93 | 1.38 | 0.205 | |

P-value

* P < 0.10

** P < 0.05

*** P < 0.01

Ref: Reference Category

biases of health workers providing less favorable contraceptives services to young, unmarried sexually active women [8, 11]. While fewer in number, the ramifications of contraceptive access are equally critical for these women, their families, and community.

Beyond Kenya, analysis of unmarried women with less recent sexual intercourse is limited or nonexistent; therefore, it is unclear whether these differences exhibited between female subgroups in Kenya are common in SSA or an aberration. Conducting analyses like this study for other countries or regions is an important topic for future research. Moreover, implementation research is needed on how to design family planning programs for these marginalized female subgroups, who already experience stigma within the health delivery system [8, 11], without limiting the reproductive health improvements for all women. Building a research agenda to better understand unmarried sexually active women can help counteract the systemic bias against these women that pervades all levels of the global healthcare system.

**Table 6. Logistic regression model for unmet need by survey round, PMA/Kenya (2014, 2019).**

| Characteristic | Categories | KENYA 2014 (Baseline) OR (95% CI) | | | P-value | | KENYA 2019 (Endline) OR (95% CI) | | | P-value | |
|---|---|---|---|---|---|---|---|---|---|---|---|
| | | Odds Ratio | Lower bound | Upper bound | | | Odds Ratio | Lower bound | Upper bound | | |
| **Age Group** | 15–24 (ref) | 1 | | | | | 1 | | | | |
| | 25–34 | 0.88 | 0.62 | 1.25 | 0.479 | | 0.68 | 0.55 | 0.85 | 0.001 | *** |
| | 35–44 | 1.12 | 0.76 | 1.65 | 0.558 | | 0.62 | 0.48 | 0.81 | 0.001 | *** |
| | 45–49 | 0.90 | 0.51 | 1.60 | 0.714 | | 0.37 | 0.25 | 0.54 | 0.000 | *** |
| **Parity** | None (ref) | 1 | | | | | 1 | | | | |
| | 1–2 | 0.91 | 0.62 | 1.33 | 0.620 | | 0.79 | 0.61 | 1.01 | 0.055 | * |
| | 3–4 | 0.93 | 0.62 | 1.41 | 0.741 | | 0.96 | 0.68 | 1.35 | 0.801 | |
| | 5 plus | 1.28 | 0.75 | 2.21 | 0.361 | | 1.39 | 0.96 | 2.01 | 0.078 | * |
| **Education** | Never (ref) | 1 | | | | | 1 | | | | |
| | Primary | 0.35 | 0.21 | 0.56 | 0.000 | *** | 0.66 | 0.45 | 0.99 | 0.031 | ** |
| | Secondary or more | 0.36 | 0.22 | 0.58 | 0.000 | *** | 0.61 | 0.40 | 0.93 | 0.022 | ** |
| **Household wealth quintile** | Lowest (ref) | 1 | | | | | 1 | | | | |
| | Middle lowest | 0.69 | 0.52 | 0.91 | 0.009 | *** | 0.99 | 0.77 | 1.26 | 0.906 | |
| | Middle | 0.57 | 0.41 | 0.80 | 0.002 | *** | 0.88 | 0.68 | 1.14 | 0.325 | |
| | Middle highest | 0.65 | 0.44 | 0.95 | 0.026 | ** | 0.72 | 0.54 | 0.95 | 0.022 | ** |
| | Highest | 0.56 | 0.35 | 0.89 | 0.016 | ** | 0.70 | 0.49 | 1.01 | 0.057 | * |
| **Residence** | Urban (ref) | 1 | | | | | 1 | | | | |
| | Rural | 1.04 | 0.72 | 1.50 | 0.920 | | 1.30 | 1.01 | 1.67 | 0.040 | ** |
| **Female subgroup** | Unmarried sexually active (1–12 months) | 1 | | | | | 1 | | | | |
| | Unmarried sexually active (0–30 days) | 13.47 | 6.30 | 28.83 | 0.000 | *** | 2.01 | 1.29 | 3.13 | 0.002 | *** |
| | Married or in union | 5.12 | 2.71 | 9.65 | 0.000 | *** | 1.16 | 0.81 | 1.66 | 0.416 | |
| **Fees for health provider** | No (ref) | 1 | | | | | 1 | | | | |
| | Yes | 1.11 | 0.82 | 1.50 | 0.496 | | 0.79 | 0.53 | 1.19 | 0.265 | |
| **Recently heard FP message** | No (ref) | 1 | | | | | 1 | | | | |
| | Yes | 0.83 | 0.57 | 1.19 | 0.304 | | 0.84 | 0.68 | 1.05 | 0.127 | |

P-value

* P < 0.10

** P < 0.05

*** P <0.01

Ref: Reference Category

This analysis has unique strengths, such as minimizing temporal bias by matching the time intervals (i.e., 12 months prior to survey) for recent ECP use and sexual activity of unmarried women; however, several limitations must be identified. First, as previously mentioned, there is a timing misalignment between 'sexually active' in last 12 months and the calculation of 'current' use of contraceptives, which can underestimate contraceptive use and increase unmet need when extending the sexual activity time interval [31]. Analysis of mCPR at last sexual intercourse was incorporated to mitigate this limitation and strengthen interpretation of the results. Second, there is a relatively low sample size for the unmarried groups, particularly in the early surveys, which limits statistical power. Third, there are slight differences in the survey questionnaires between survey years, such as no question on ECP in last 12 months within the 2014 questionnaire. Fourth, the datasets provided by PMA do not allow for linking the

**Table 7. Logistic regression model for recent emergency contraceptive use (in last 12 months) by survey round, PMA/Kenya (2017, 2019).**

| Characteristic | Categories | KENYA 2017 (Baseline) | | | | | KENYA 2019 (Endline) | | | | |
|---|---|---|---|---|---|---|---|---|---|---|---|
| | | OR (95% CI) | | | P-value | | OR (95% CI) | | | P-value | |
| | | Odds Ratio | Lower bound | Upper bound | | | Odds Ratio | Lower bound | Upper bound | | |
| **Age Group** | 15–24 (ref) | 1 | | | | | 1 | | | | |
| | 25–34 | 1.55 | 1.00 | 2.40 | 0.049 | ** | 1.04 | 0.78 | 1.39 | 0.783 | |
| | 35–44 | 1.07 | 0.53 | 2.14 | 0.850 | | 0.85 | 0.59 | 1.23 | 0.391 | |
| | 45–49 | 0.48 | 0.14 | 1.61 | 0.230 | | 0.42 | 0.22 | 0.79 | 0.008 | *** |
| **Parity** | None (ref) | 1 | | | | | 1 | | | | |
| | 1–2 | 0.47 | 0.31 | 0.71 | 0.000 | *** | 0.80 | 0.53 | 1.21 | 0.291 | |
| | 3–4 | 0.65 | 0.41 | 1.02 | 0.063 | * | 0.83 | 0.52 | 1.33 | 0.430 | |
| | 5 plus | 0.58 | 0.24 | 1.39 | 0.216 | | 0.86 | 0.47 | 1.57 | 0.614 | |
| **Education** | Never (ref) | 1 | | | | | 1 | | | | |
| | Primary | 0.48 | 0.34 | 0.68 | 0.002 | *** | 1.42 | 0.48 | 4.19 | 0.526 | |
| | Secondary or more | 1.00 | – | – | – | | 2.60 | 0.83 | 8.19 | 0.102 | |
| **Household wealth quintile** | Lowest (ref) | 1 | | | | | 1 | | | | |
| | Middle lowest | 1.66 | 0.88 | 3.11 | 0.114 | | 1.11 | 0.66 | 1.86 | 0.697 | |
| | Middle | 2.08 | 0.98 | 4.41 | 0.056 | * | 1.20 | 0.72 | 1.99 | 0.481 | |
| | Middle highest | 1.53 | 0.67 | 3.49 | 0.314 | | 1.89 | 1.14 | 3.14 | 0.014 | ** |
| | Highest | 2.64 | 1.10 | 6.31 | 0.030 | ** | 1.95 | 1.14 | 3.31 | 0.015 | ** |
| **Residence** | Urban (ref) | 1 | | | | | 1 | | | | |
| | Rural | 1.13 | 0.54 | 2.35 | 0.749 | | 0.77 | 0.57 | 1.04 | 0.091 | * |
| **Female subgroup** | Unmarried sexually active (1–12 months) | 1 | | | | | 1 | | | | |
| | Unmarried sexually active (0–30 days) | 1.53 | 0.85 | 2.74 | 0.150 | | 0.93 | 0.60 | 1.43 | 0.727 | |
| | Married or in union | 0.71 | 0.43 | 1.18 | 0.184 | | 0.60 | 0.44 | 0.82 | 0.001 | *** |
| **Fees for health provider** | No (ref) | 1 | | | | | 1 | | | | |
| | Yes | 0.56 | 0.23 | 1.36 | 0.198 | | 1.13 | 0.74 | 1.71 | 0.577 | |
| **Recently heard FP message** | No (ref) | 1 | | | | | 1 | | | | |
| | Yes | 1.13 | 0.58 | 2.21 | 0.711 | | 1.00 | 0.67 | 1.50 | 0.991 | |

P-value

\* P < 0.10

\*\* P < 0.05

\*\*\* P <0.01

Ref: Reference Category

respondent to the SDP of actual service delivery, which lessens the validity of service delivery indicators. Also, unmarried women are more likely to underreport sexual activity and contraceptive use, but research to quantify this bias is limited [8]. Lastly, analysis of women in Kenya may not be emblematic or representative of other countries or regions in SSA.

In Kenya, contraceptive use and unmet need are asymmetrically changing among female subgroups. Unmarried women with less recent sexual activity exhibit different modern contraceptive use and unmet need; however, reporting and research by the measurement community is extremely limited for these women. Recognizing this measurement bias and generating targeted information for these marginalized groups is the first step towards inclusive decision-making and equitable service delivery.

## Supporting information

**S1 Appendix.**
(DOCX)

## Author Contributions

**Conceptualization:** Bennett Nemser.

**Data curation:** Bennett Nemser, Nicholas Addofoh.

**Formal analysis:** Nicholas Addofoh.

**Investigation:** Bennett Nemser.

**Methodology:** Bennett Nemser, Nicholas Addofoh.

**Project administration:** Bennett Nemser.

**Supervision:** Bennett Nemser.

**Validation:** Bennett Nemser, Nicholas Addofoh.

**Visualization:** Bennett Nemser.

**Writing – original draft:** Bennett Nemser.

**Writing – review & editing:** Bennett Nemser, Nicholas Addofoh.

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
