## [Decision Letter · Decision Letter 0]

3 Jan 2022

PONE-D-21-35162Contraceptive Demand and Utilization by Unmarried, Sexually Active Women in Kenya: A Multilevel Regression AnalysisPLOS ONE

Dear Dr. Nemser,

Thank you for submitting your manuscript to PLOS ONE. After careful consideration, we feel that it has merit but does not fully meet PLOS ONE’s publication criteria as it currently stands. Therefore, we invite you to submit a revised version of the manuscript that addresses the points raised during the review process.

We look forward to receiving your revised manuscript.

Kind regards,

Joseph KB Matovu, Ph.D.

Academic Editor

PLOS ONE

Reviewers' comments:

Reviewer's Responses to Questions

**Comments to the Author**

1. Is the manuscript technically sound, and do the data support the conclusions?

Reviewer #1: Yes

Reviewer #2: Yes

Reviewer #3: Partly

2. Has the statistical analysis been performed appropriately and rigorously? 

Reviewer #1: Yes

Reviewer #2: N/A

Reviewer #3: No

3. Have the authors made all data underlying the findings in their manuscript fully available?

Reviewer #1: Yes

Reviewer #2: Yes

Reviewer #3: Yes

4. Is the manuscript presented in an intelligible fashion and written in standard English?

Reviewer #1: Yes

Reviewer #2: Yes

Reviewer #3: No

5. Review Comments to the Author

Reviewer #1: This is an interesting and generally well written manuscript describing contraceptive demand and utilization in unmarried sexually active women in Kenya. I believe that the manuscript would benefit from some revisions before publication.

Suggested revisions below:

Abstract

- Although the text is generally clear, there are parts where it is very biostats language heavy and could benefit from edits to improve clarity regarding the clinical implications for example "trending in different directions", "survey enumeration" in the background section. Whilst I appreciate that these terms are correct, it may be clearer to say "differences particularly in women not sexually active 1-12 months", or "reporting no recent sexual activity in preceding month".

Background

- It would be helpful to get additional context regarding the Kenyan population. From the manuscript it appears that most women are married, but I'm not certain if this is true generally in Kenya or whether this is true for women who contributed to the data. This would be helpful to understand. This could potentially provide substantial bias as there may be stigma associated with sexual activity outside of marriage and mainly unmarried women may be less likely to contribute. Similarly, there is an implication that unmarried women do not have sexual intercourse which needs to be clarified.

- Do women in Kenya ever use the copper IUD as EC? Only the contraceptive pill EC is discussed.

- In the paragraph: "With a population of ......" in the 4th line it is difficult to understand the breakdown of the 39% of women using contraception (CPR) please edit for clarity

- In the last paragraph "For family planning indicators....." a number of sentences would fit better into methods-suggest removing these and focussing on how this study addresses outstanding issues.

- The last paragraph of the background: additional text here can also be moved to methods

Methods

- Generally clear

Results

- The results are interesting and well represented

- It is particularly interesting that women who are in the unmarried sexually active group are the highest users of EC. In the discussion this is posed as a negative, but perhaps the authors should also emphasise that this is a real positive in terms of women perhaps choosing not to permanently use contraception, but able to access EC when needed. Obviously a longer term choice may be preferable but it is important that women are able to access this option

- Is there a correlation between the sexually active women with unmet contraception need and pregnancy in this study? If women are not becoming pregnant unplanned, then perhaps this is not as much of a concern.

- Also interesting the increase in implant use in this cohort

- The tables are good and clear

- The figures are difficult to understand as no figure headings and legends are included for figure 2 onwards

Discussion

- Well written

- Please see previous comments regarding clarifying this issues and findings, for example the fact that unmarried sexually active women 1-12 months access EC is a plus rather than concerning in my opinion, hopefully averting unplanned pregnancies.

- I'm not sure that the 3rd paragraph is substantiated. "The female subpopulations....". The paper doesn't report pregnancy in the group of unmarried sexually active, so it is possible that using EC is acceptable particularly if the encounter was unplanned/unexpected. It is a positive that women have sufficient agency to choose when they wish to access contraception. Obviously in this cohort if women do have unplanned pregnancies then this is a concern.

- In the paragraph " To reach universal...." whilst it is appreciated that unmarried women who are sexually active 1-12 months may be of concern, I wonder how relevant this specific focus on marriage is, to many women in different countries and regions. There is a danger of focussing separately on this group as this could be stigmatising and essentially messages regarding contraceptive access should reach all women. I realise that is what the authors are saying and perhaps more context regarding marriage in Kenya would be helpful. It seems unlikely that there are so few unmarried sexually active women relative to married women and in itself it seems to perpetuate the message that unmarried women should not be sexually active, which is problematic and likely compounds the issue. Perhaps these points can be discussed.

Reviewer #2: Thank you for the opportunity to review this article. I would only suggest a few changes in this paragraph (2nd para. of Introduction):

(…) The emergency contraceptive pill (EC) is an oral, hormonal contraceptive pill for women to use as soon as possible (up to 5 days) after sexual intercourse to prevent unwanted pregnancy. EC can help prevent pregnancies due to non-use, failure or misuse of

contraceptive, or situations of rape or coerced sex10,11. EC has a pregnancy prevention rate ranging from 56% to 95% if promptly and appropriately administered12–16. Suitably, EC was selected as one of 13 high impact, low-cost commodities by the UN Commission on Life-saving Commodities for Women and Children (UNCoLSC)17. EC use is highest among two groups of women: aged 20-24 years and unmarried sexually active18,19. EC is safe for over-the-counter sale and often available from a pharmacist or drug seller without a prescription20.

1. Remove “hormonal”. Currently there are two types of oral emergency contraption (EC) pills more widely used: one contains levonorgestrel (LNG) which is a hormone; but the other one contains ulipristal acetate (UPA), which is a selective progesterone receptor modulator (SPRM).

2. Emergency contraception refers to pills but also to the use of the IUD. Since the article seems to refer to EC pills I would make it explicit (talk about “emergency contraception pills”)

3. I would refer to “pills” in plural.

4. It is not clear to me what “appropriately administered” refers to. I would rather say “if promptly used”.

5. For reference, WHO’s factsheet on EC can provide further clarity: https://www.who.int/news-room/fact-sheets/detail/emergency-contraception

I hope this is useful. Thank you for this effort to make more visible important subgroups of population that are chronically underrepresented in global health measurements. For the EC community I think the point the paper makes is very valuable too. Congratulations!

Reviewer #3: Review Outcome

Title:

• Suggest the title to be modified

• Suggested title: Trends in Contraceptive Demand and Utilization Among Sexually Active Unmarried Women in Kenya: A Multilevel Regression Analysis

Abstract

• Authors have used the term demand and need interchangeably

• The first statement in the background sub-section is not clear***require modification

• If the primary aim of this paper is to analyze trends in contraceptive utilization and demand, the title should be modified accordingly, as suggested above

• Grammatical error: In the methods, authors stated as: ‘This study analyzed datasets*****’……datasets can’t be analyzed***revise the statements to make it clear and concise.

• Results seems a conclusion statement. Authors should incorporate regression results and 95% CI. They should also describe sample size included in the analysis.

• Conclusions should be drawn based on the aim of the study. No statement in the conclusion referred to trends in contraceptive demand and utilization, and associated consequences

• Finally, authors should avoid using abbreviations

• In general, abstract is not informative and require intense grammar revisions.

Introduction

• Authors have tried to synthesize contraceptive demand and use in the global, regional and study area context including the consequences of non-utilization.

• Authors should revise language***with some statements lacked clarity and coherence of ideas. For instance: ‘Even with this comparatively high performance, Kenya implemented policies to

reduce barriers to access family planning, such as policies enacted in 2013 to effectively eliminate family planning user fees as well as other public outpatient costs

• Authors should discuss the approach or methods followed including data sources in the methods section. For instance: ‘This analysis utilized data from the Performance Monitoring and Accountability (PMA) survey25. Managed by the Kenya Ministry of Health, PMA was a nationally representative survey of female respondents along with service delivery points (e.g., health facilities) to understand family planning usage, knowledge, and experience of women as well as service availability in the community. In addition, PMA incorporated a unique EC question: “Have you used emergency contraception at any time in the last 12 months?”. This question has a longer recall period than the traditional ‘current use’ EC indicator, which underestimates the scale of EC usage’

• The last statement of the introduction (aim statement) is not clear. Authors should clearly specify the aim of the study and aim should be consistent to the one stated in the abstract section and the title of the paper.

• Moreover, authors should conduct language and grammar revisions (for instance: check the 2nd paragraph)

• The conceptual framework should be presented as part of methods (Figure 1)

Methods

• Methods lacks clarity. Authors should clearly describe the method they have used to answer the aim or research question. Suggested sub-sections can be:

o Study setting

o Data source and measurements

o Study variables

o Statistical analysis

• Authors should avoid use of some jargon or non-technical words. For instance: ‘The female questionnaire includes marital status, recency of sexual activity, *****’

• Tables should be self-explanatory, with proper footnote and need to be properly cited inside the document.

• Authors shouldn’t include variable definition as supplementary file.

• The analysis methods used is not clear. Authors should clarify, why and how they have used the multi-level regression model.

Results

• Table/figure titles should be self-explanatory and tables need to be properly cited within the document. Avoid citing like, (see Table 2; see Figure 2); rather (Table 2; Figure 2). For each table, the source of data should be indicated.

• What is the need to include ‘change’ in Table 3? Try to use the proper color whenever presenting figures.

• In Table 3, for each year, include both number and %. All abbreviations should be described as foot note. What do, other traditional included?

• This section is not clear and difficult to follow-up. Authors should organize and briefly present the findings based on objectives of the analysis.

• Table 4.1-4.4 is not clear. What is the aim of this analysis?

• In general, authors should answer previously formulated research questions.

Discussion:

• Authors should properly discuss the theoretical and practical implications of the analysis

• They should adequately discuss the findings in the context of other settings

• Strength and limitations of the analysis need to be explained

• The conclusion should be based on the findings of the analysis

Final Decision: Reject

6. PLOS authors have the option to publish the peer review history of their article (what does this mean?). If published, this will include your full peer review and any attached files.

Reviewer #1: **Yes: **Lee Fairlie

Reviewer #2: **Yes: **Cristina Puig Borràs

Reviewer #3: No

---

## [Author Response · Author response to Decision Letter 0]

18 Feb 2022

5. Review Comments to the Author

Reviewer #1: This is an interesting and generally well written manuscript describing contraceptive demand and utilization in unmarried sexually active women in Kenya. I believe that the manuscript would benefit from some revisions before publication.

Suggested revisions below:

Abstract

- Although the text is generally clear, there are parts where it is very biostats language heavy and could benefit from edits to improve clarity regarding the clinical implications for example "trending in different directions", "survey enumeration" in the background section. Whilst I appreciate that these terms are correct, it may be clearer to say "differences particularly in women not sexually active 1-12 months", or "reporting no recent sexual activity in preceding month".

Background

- It would be helpful to get additional context regarding the Kenyan population. From the manuscript it appears that most women are married, but I'm not certain if this is true generally in Kenya or whether this is true for women who contributed to the data. This would be helpful to understand. This could potentially provide substantial bias as there may be stigma associated with sexual activity outside of marriage and mainly unmarried women may be less likely to contribute. Similarly, there is an implication that unmarried women do not have sexual intercourse which needs to be clarified.

Authors. This is a good point. We added more background details on the percentage of married women in Kenya (60%) to the third Introduction paragraph and clarified the rates of sexual activity in the fourth paragraph, where 6.8% of unmarried women were sexually active (in last month) in Kenya. Our sample has about 9% of unmarried women who are sexually active (in last month) and another 14% sexually active (1-12 months). 

- Do women in Kenya ever use the copper IUD as EC? Only the contraceptive pill EC is discussed.

Authors: Thank you for the comment. We have added a comment on copper IUD to the Introduction and clarified the focus on emergency contraceptive pills. 

- In the paragraph: "With a population of ......" in the 4th line it is difficult to understand the breakdown of the 39% of women using contraception (CPR) please edit for clarity

Authors: Thank you. We edited the paragraph for added clarity. 

- In the last paragraph "For family planning indicators....." a number of sentences would fit better into methods-suggest removing these and focussing on how this study addresses outstanding issues.

Authors: We appreciate the suggestion. We moved the discussion on temporal misalignment to the Methods section. 

- The last paragraph of the background: additional text here can also be moved to methods

Authors: Thank you. We removed one of the sentences for clarity. 

Methods

- Generally clear

Results

- The results are interesting and well represented

- It is particularly interesting that women who are in the unmarried sexually active group are the highest users of EC. In the discussion this is posed as a negative, but perhaps the authors should also emphasise that this is a real positive in terms of women perhaps choosing not to permanently use contraception, but able to access EC when needed. Obviously a longer term choice may be preferable but it is important that women are able to access this option

Authors: Thank you for the comment. This was not our intention – we tried to remain neutral by using words such as “shifting”. We edited the last sentence in that paragraph to illustrate that the women are ‘leveraging’ access to ECP to reduce unwanted pregnancies. 

- Is there a correlation between the sexually active women with unmet contraception need and pregnancy in this study? If women are not becoming pregnant unplanned, then perhaps this is not as much of a concern.

Authors: This is an interesting research question and something the authors considered, but unfortunately the PMA questionnaire is not well designed to answer and beyond the scope of this study. 

- Also interesting the increase in implant use in this cohort

- The tables are good and clear

- The figures are difficult to understand as no figure headings and legends are included for figure 2 onwards

Authors: Apologies, this was an upload error and all tables / figures have now been properly titled and cited. 

Discussion

- Well written

- Please see previous comments regarding clarifying this issues and findings, for example the fact that unmarried sexually active women 1-12 months access EC is a plus rather than concerning in my opinion, hopefully averting unplanned pregnancies.

Authors: See our comment above. 

- I'm not sure that the 3rd paragraph is substantiated. "The female subpopulations....". The paper doesn't report pregnancy in the group of unmarried sexually active, so it is possible that using EC is acceptable particularly if the encounter was unplanned/unexpected. It is a positive that women have sufficient agency to choose when they wish to access contraception. Obviously in this cohort if women do have unplanned pregnancies then this is a concern.

Authors: Very good point. We added additional commentary on this issue – a decrease in unmet need for ‘current’ mCPR, may indicate women have more agency to access ECP. It is certainly an area for future research. 

- In the paragraph " To reach universal...." whilst it is appreciated that unmarried women who are sexually active 1-12 months may be of concern, I wonder how relevant this specific focus on marriage is, to many women in different countries and regions. There is a danger of focussing separately on this group as this could be stigmatising and essentially messages regarding contraceptive access should reach all women. I realise that is what the authors are saying and perhaps more context regarding marriage in Kenya would be helpful. It seems unlikely that there are so few unmarried sexually active women relative to married women and in itself it seems to perpetuate the message that unmarried women should not be sexually active, which is problematic and likely compounds the issue. Perhaps these points can be discussed.

Authors: Thank you for the comment. In short, 13% of the women in the study were unmarried and sexually active 1-12 months prior to survey, so it’s a sizable subgroup. Your question focuses on the ‘how’ to address contraceptive disparities in these subgroups (e.g., without using messaging that further stigmatizes them). The authors felt this type of discussion was complex and nuanced and beyond the scope this section. But, identifying the problem is the first step to a solution. In addition, we added this sentence to the discussion to help address this issue: “Moreover, implementation research is needed on how to design family planning programs for these marginalized female subgroups, who already experience stigma within the health delivery system, without limiting the reproductive health improvements for all women.” 

Reviewer #2: Thank you for the opportunity to review this article. I would only suggest a few changes in this paragraph (2nd para. of Introduction):

(…) The emergency contraceptive pill (EC) is an oral, hormonal contraceptive pill for women to use as soon as possible (up to 5 days) after sexual intercourse to prevent unwanted pregnancy. EC can help prevent pregnancies due to non-use, failure or misuse of

contraceptive, or situations of rape or coerced sex10,11. EC has a pregnancy prevention rate ranging from 56% to 95% if promptly and appropriately administered12–16. Suitably, EC was selected as one of 13 high impact, low-cost commodities by the UN Commission on Life-saving Commodities for Women and Children (UNCoLSC)17. EC use is highest among two groups of women: aged 20-24 years and unmarried sexually active18,19. EC is safe for over-the-counter sale and often available from a pharmacist or drug seller without a prescription20.

1. Remove “hormonal”. Currently there are two types of oral emergency contraption (EC) pills more widely used: one contains levonorgestrel (LNG) which is a hormone; but the other one contains ulipristal acetate (UPA), which is a selective progesterone receptor modulator (SPRM).

Authors: Thank you for the feedback. This has been edited. 

2. Emergency contraception refers to pills but also to the use of the IUD. Since the article seems to refer to EC pills I would make it explicit (talk about “emergency contraception pills”)

Authors: Thank you for the comment. This has been edited. 

3. I would refer to “pills” in plural.

Authors: Thank you for the feedback. This has been updated. 

4. It is not clear to me what “appropriately administered” refers to. I would rather say “if promptly used”.

Authors: Thank you for the feedback. This has been edited. 

5. For reference, WHO’s factsheet on EC can provide further clarity: https://www.who.int/news-room/fact-sheets/detail/emergency-contraception

I hope this is useful. Thank you for this effort to make more visible important subgroups of population that are chronically underrepresented in global health measurements. For the EC community I think the point the paper makes is very valuable too. Congratulations!

Reviewer #3: Review Outcome

Title:

• Suggest the title to be modified

• Suggested title: Trends in Contraceptive Demand and Utilization Among Sexually Active Unmarried Women in Kenya: A Multilevel Regression Analysis

Authors: Thank you for the feedback. The title has been edited. 

Abstract

• Authors have used the term demand and need interchangeably

Authors: Thank you for pointing this out. We reviewed the entire document and updated terminology to ensure clarity between these two terms. 

 • The first statement in the background sub-section is not clear***require modification

Authors: Thank you for the feedback. This has been modified. 

• If the primary aim of this paper is to analyze trends in contraceptive utilization and demand, the title should be modified accordingly, as suggested above

Authors: Thank you. The title has been modified. 

• Grammatical error: In the methods, authors stated as: ‘This study analyzed datasets*****’……datasets can’t be analyzed***revise the statements to make it clear and concise.

Authors: Thank you for identifying this oversight. It has been edited. 

• Results seems a conclusion statement. Authors should incorporate regression results and 95% CI. They should also describe sample size included in the analysis.

Authors: Thank you for raising this important point. We added the sample size to the Abstract as well as the regression results. The Abstract length – particularly the Results sub-section - was expanded to accommodate these important points. Initially, we tried to keep the Abstract very short, but we unfortunately it appears we sacrificed clarity and adequate understanding by the reader. Thank you for pointing this out. 

• Conclusions should be drawn based on the aim of the study. No statement in the conclusion referred to trends in contraceptive demand and utilization, and associated consequences

Authors: Thank you for this feedback. We restructured the concluding paragraph. The first two sentences refer to the aims of the study. The last two sentences address the implications and forward-looking research needs. 

• Finally, authors should avoid using abbreviations

Authors: Yes, we were conscious of this and tried to limit abbreviations as much as possible. However, we included two common abbreviations mCPR and ECP, which were defined and repeated multiple times in the Abstract, to reduce the word count. This is consistent with other PLOS One manuscripts, such as Shiferaw 2017, which is referenced in this manuscript. 

• In general, abstract is not informative and require intense grammar revisions.

Authors: Thank you again for this feedback. We conducted a comprehensive review of the Abstract and made large-scale edits to increase clarity and information transfer. 

Introduction

• Authors have tried to synthesize contraceptive demand and use in the global, regional and study area context including the consequences of non-utilization.

• Authors should revise language***with some statements lacked clarity and coherence of ideas. For instance: ‘Even with this comparatively high performance, Kenya implemented policies to

reduce barriers to access family planning, such as policies enacted in 2013 to effectively eliminate family planning user fees as well as other public outpatient costs

Authors: Thank you for the feedback. This has been edited. 

• Authors should discuss the approach or methods followed including data sources in the methods section. For instance: ‘This analysis utilized data from the Performance Monitoring and Accountability (PMA) survey25. Managed by the Kenya Ministry of Health, PMA was a nationally representative survey of female respondents along with service delivery points (e.g., health facilities) to understand family planning usage, knowledge, and experience of women as well as service availability in the community. In addition, PMA incorporated a unique EC question: “Have you used emergency contraception at any time in the last 12 months?”. This question has a longer recall period than the traditional ‘current use’ EC indicator, which underestimates the scale of EC usage’

Authors: Thank you for the feedback. We shortened this section, but we need to include some mention of PMA and the emergency contraception question in the Introduction, because it provides background for the Study Aims in the last paragraph of the Introduction. 

• The last statement of the introduction (aim statement) is not clear. Authors should clearly specify the aim of the study and aim should be consistent to the one stated in the abstract section and the title of the paper.

Authors: Thank you. We updated and simplified the aims statement.

• Moreover, authors should conduct language and grammar revisions (for instance: check the 2nd paragraph)

Authors: Thank you for the recommendation. We have made significant content and grammatic revisions to the second paragraph of the Introduction on ECP. We added more 

• The conceptual framework should be presented as part of methods (Figure 1)

Authors: Thank you for the feedback. The conceptual framework has been moved to the Methods section. 

Methods

• Methods lacks clarity. Authors should clearly describe the method they have used to answer the aim or research question. Suggested sub-sections can be:

o Study setting

o Data source and measurements

o Study variables

o Statistical analysis

Authors: Thank you for the feedback. The sub-headings have been edited. 

• Authors should avoid use of some jargon or non-technical words. For instance: ‘The female questionnaire includes marital status, recency of sexual activity, *****’

Authors: Thank you for the comment. This sentence has been simplified. 

• Tables should be self-explanatory, with proper footnote and need to be properly cited inside the document.

Authors: Thank you for the comment. This sentence has been updated. 

• Authors shouldn’t include variable definition as supplementary file.

Authors: Thank you for the comment, but we are unclear as to the corrective action. We are happy to remove the variable definition file altogether or move it into the main body of the manuscript. If PLOS One can provide guidance on standard format, that would be helpful. 

• The analysis methods used is not clear. Authors should clarify, why and how they have used the multi-level regression model.

Authors: Thank you for the comment. We have added additional description on why and how the multi-level regression models were used. 

Results

• Table/figure titles should be self-explanatory and tables need to be properly cited within the document. Avoid citing like, (see Table 2; see Figure 2); rather (Table 2; Figure 2). For each table, the source of data should be indicated.

Authors: Apologies, this was an upload error and all tables / figures have now been properly titled and cited. 

• What is the need to include ‘change’ in Table 3? Try to use the proper color whenever presenting figures .

Authors: Thank you for the comment. This is simply a visual aid for the reader. The table includes about 200 figures, so the authors felt a quick reference for the reader was advantageous. 

• In Table 3, for each year, include both number and %. All abbreviations should be described as foot note. What do, other traditional included?

Authors: Thank you for the feedback. We added explanations of abbreviations and definitions including for ‘other traditional’ methods. We added the total number of women for each column, which is consistent with standard presentation of method mix in other sources, such as DHS. 

• This section is not clear and difficult to follow-up. Authors should organize and briefly present the findings based on objectives of the analysis.

Authors: Thank you for the comment. We clarified the Aims Statement and utilized similar subheaders for the Results section to make the connection more clear. 

• Table 4.1-4.4 is not clear. What is the aim of this analysis?

Authors: Thank you for the feedback. We updated the Table titles, narrative and clarified the Aims Statement to more directly link with these Tables and Results section. Hopefully, these changes make the link and utility of the tables more clear. 

• In general, authors should answer previously formulated research questions.

Authors: The research questions have been more clearly stated in the Aims Statement and matches the flow of the Results section. The three areas of research are:

1. Level and trend of contraceptive use and unmet need indicators (mCPR, unmet need, demand, etc) by female subgroup.

2. Level and trend of recent ECP use by female subgroup

3. Influence of contextual factors (e.g., demographic, socioeconomic, female subgroup) on the outcome measures - over time 

Discussion:

• Authors should properly discuss the theoretical and practical implications of the analysis

Authors: Thank you for mentioning this. It was a catalyst for a deeper review of the Discussion and make it more clear to the reader. We added several sentences (2nd, 3rd, 4th and 5th paragraphs) to more clearly identify the theoretical and practical implications of this analysis. 

• They should adequately discuss the findings in the context of other settings

Authors: Thank you – this was useful feedback. Data is limited internationally, so it is hard to make concrete assessments in other settings. We decided to add a paragraph to the Discussion (6th paragraph) to discuss this important issue:

“Beyond Kenya, analysis of unmarried women with less recent sexual intercourse is limited or nonexistent; therefore, it is unclear whether these differences exhibited between female subgroups in Kenya are common in SSA or an aberration. Conducting analyses like this study for other countries or regions is an important topic for future research. Moreover, implementation research is needed on how to design family planning programs for these marginalized female subgroups, who already experience stigma within the health delivery system, without limiting the reproductive health improvements for all women. Building a research agenda to better understand unmarried sexually active women can help counteract the systemic bias against these women that pervades all levels of the global healthcare system.” 

• Strength and limitations of the analysis need to be explained

Authors: Thank you for this comment. The 7th paragraph includes the limitations and we added more context about the strength of the analysis here (that wasn’t covered elsewhere in the manuscript, such as the Methods and earlier in Discussion). 

• The conclusion should be based on the findings of the analysis

Authors: Thank you for this feedback. We shortened the concluding paragraph. The first two sentences summarize the findings in layperson’s terms, while the last sentence is a more forward looking perspective.

---

## [Decision Letter · Decision Letter 1]

20 Apr 2022

PONE-D-21-35162R1Trends and Contextual Factors associated with Contraceptive Utilization and Unmet Need Among Sexually Active Unmarried Women in Kenya: A Multilevel Regression AnalysisPLOS ONE

Dear Dr. Nemser:

Thank you for submitting your manuscript to PLOS ONE. After careful consideration, we feel that it has merit but does not fully meet PLOS ONE’s publication criteria as it currently stands. Therefore, we invite you to submit a revised version of the manuscript that addresses the points raised during the review process. Abstract & background

1. In the background section of the manuscript, the authors write: “This study aims to explore the level and trend of contraceptive use and unmet family planning needs among female subpopulations in Kenya: married or in union (i.e., living together); unmarried and sexually active within the past 30 days prior to survey (labeled as UA-30days); and unmarried and sexually active between 1-12 months prior to the survey (labeled as UA-12months)”. In the abstract, the authors write: “Unmarried women who report less recent sexual intercourse (>30 days from survey enumeration) are largely excluded from global health monitoring and evaluation efforts”. A few points to note here:

*In the statement of the objective picked from the background section, the authors refer to ‘… the level and trend’ of contraceptive us – but the reference to ‘level’ is not mentioned in the statement of the objective picked from the abstract. The statement of the overall objective should be consistently presented across the manuscript

*The statement of the objective picked from the background section uses the verb ‘explore’. I suggest that this verb be revised to a more quantitative term that explains what the authors actually did.

*In the statement of the objective picked from the background section, the authors refer to ‘female subpopulations in Kenya’ with a list of these sub-populations listed to include: married or in union, unmarried and sexually active within the past 30 days prior to survey and unmarried and sexually active between 1-12 months prior to the survey. Based on these subgroups, I wonder if the use of only ‘unmarried women’ in the title is appropriate.

*In the abstract, the authors focus on ‘unmarried women who report less recent sexual intercourse’ but the statement of the objective in the background section refers to three subpopulations. Can the authors clarify on why this is the case?

2. If the aim of the study was to assess “trends in … contraceptive utilization and unmet need”, then, I would expect to see these trends presented in the abstract. The authors indicate that they used data collected in multiple surveys over the period 2014 to 2019; so, I expected to see some trend analyses presented, and I would be interested to know if there was a significant increasing or decreasing trend or whether there was no change over the years in contraceptive utilization and unmet need. This is not provided. Instead, the authors focus on reporting on 2019 indicators which makes it difficult to tell if the analysis was to assess trends or just contraceptive utilization in 2019. Also, the reporting on the trends in unmet need comes at the extreme end of the results sub-section, and presented in a more generic format.

3. I realize that the issue of emergency contraception is singled out in the abstract. Why was this singled out and not considered as one of the contraceptive methods, in the same way the other short-term methods were handled.

Results

1. Table 3 presents the contraceptive method mix by female Subgroup for the three surveys considered in the analysis (2014, 2017, 2019). Did the authors try to assess if the observed changes, as reported, depicted significant increases or decreases in the outcome over time? This question also applies to Figure 4 on unmet need.

We look forward to receiving your revised manuscript.

Kind regards,

Joseph KB Matovu, Ph.D.

Academic Editor

PLOS ONE

Journal Requirements:

Additional Editor Comments (if provided):

The authors have addressed the reviewers' comments to their satisfaction and they have recommended that this manuscript be accepted for publication. However, my own review of the paper shows that there are a few areas where the authors can provide additional clarification before this paper is accepted for publication. These comments have been summarized for the authors above.

Reviewers' comments:

Reviewer's Responses to Questions

**Comments to the Author**

1. If the authors have adequately addressed your comments raised in a previous round of review and you feel that this manuscript is now acceptable for publication, you may indicate that here to bypass the “Comments to the Author” section, enter your conflict of interest statement in the “Confidential to Editor” section, and submit your "Accept" recommendation.

Reviewer #1: All comments have been addressed

Reviewer #3: All comments have been addressed

2. Is the manuscript technically sound, and do the data support the conclusions?

Reviewer #1: Yes

Reviewer #3: Yes

3. Has the statistical analysis been performed appropriately and rigorously? 

Reviewer #1: Yes

Reviewer #3: Yes

4. Have the authors made all data underlying the findings in their manuscript fully available?

Reviewer #1: Yes

Reviewer #3: Yes

5. Is the manuscript presented in an intelligible fashion and written in standard English?

Reviewer #1: Yes

Reviewer #3: Yes

6. Review Comments to the Author

Reviewer #1: Thanks you for addressing the reviewer comments. The manuscript is well written and interesting. I am happy for this manuscript to be published.

Reviewer #3: I appreciate the effort made by authors to incorporate comments that have been given previously and I have no more comments.

7. PLOS authors have the option to publish the peer review history of their article (what does this mean?). If published, this will include your full peer review and any attached files.

Reviewer #1: **Yes: **Lee Fairlie

Reviewer #3: **Yes: **Full name: Dawit Wolde Daka; http://orcid.org/0000-0001-5465-6345

---

## [Author Response · Author response to Decision Letter 1]

1 Jun 2022

Dear Dr. Nemser:

Thank you for submitting your manuscript to PLOS ONE. After careful consideration, we feel that it has merit but does not fully meet PLOS ONE’s publication criteria as it currently stands. Therefore, we invite you to submit a revised version of the manuscript that addresses the points raised during the review process.

Abstract & background

1. In the background section of the manuscript, the authors write: “This study aims to explore the level and trend of contraceptive use and unmet family planning needs among female subpopulations in Kenya: married or in union (i.e., living together); unmarried and sexually active within the past 30 days prior to survey (labeled as UA-30days); and unmarried and sexually active between 1-12 months prior to the survey (labeled as UA-12months)”. In the abstract, the authors write: “Unmarried women who report less recent sexual intercourse (>30 days from survey enumeration) are largely excluded from global health monitoring and evaluation efforts”. 

A few points to note here:

*In the statement of the objective picked from the background section, the authors refer to ‘… the level and trend’ of contraceptive us – but the reference to ‘level’ is not mentioned in the statement of the objective picked from the abstract. The statement of the overall objective should be consistently presented across the manuscript

>>AUTHORS: Thank you very much for the comment. We edited the background section to incorporate your suggestion and maintain consistency. We removed mention of evaluating trend. 

*The statement of the objective picked from the background section uses the verb ‘explore’. I suggest that this verb be revised to a more quantitative term that explains what the authors actually did.

>>AUTHORS: Thank you for the suggestion. We replaced ‘explore’ with the term ‘evaluate’ in the background section. 

*In the statement of the objective picked from the background section, the authors refer to ‘female subpopulations in Kenya’ with a list of these sub-populations listed to include: married or in union, unmarried and sexually active within the past 30 days prior to survey and unmarried and sexually active between 1-12 months prior to the survey. Based on these subgroups, I wonder if the use of only ‘unmarried women’ in the title is appropriate.

>>AUTHORS: Thank you for this feedback. As mentioned in the Introduction and Discussion, global health monitoring and evaluation efforts focus specifically on married / in union women or unmarried women with recent sexual intercourse (less than 30 days). This data processing and reporting bias has existed for decades. The data presented in this study on these two subgroups is regularly described at length in other global health reports. However, the unique feature of this analysis is the evaluation of unmarried women with less recent sexual intercourse (between 1-12 months prior to survey). The analysis of this underreported subgroup of women in this study provides the substantive contribution to public knowledge. Therefore, the title focuses on “unmarried women” to reflect the deeper investigation of these women and uniqueness and contribution of this study. 

*In the abstract, the authors focus on ‘unmarried women who report less recent sexual intercourse’ but the statement of the objective in the background section refers to three subpopulations. Can the authors clarify on why this is the case?

>>AUTHORS: Thank you for the comment. As mentioned above, the global health monitoring and evaluation community routinely (and almost exclusively) reports on married / in union women or unmarried women with recent sexual intercourse (less than 30 days). This practice is rooted in commonalities between these two sets of women relative to current mCPR indicator. As mentioned in the Methods section, the addition of ‘mCPR at last sex’ and ‘recent use of emergency contraceptive’ data collection by PMA, allows for a more intensive investigate of actual contraceptive use (unbiased by ‘current use of mCPR’) for unmarried women with less recent sexual activity (between 1-12 months prior to survey). This female subgroup makes up about 13% of the women enumerated, but they are systematically eliminated from reporting in the global health literature and thus excluded from influence on decision-making in various context. The unique aspect of this study is the focus on ‘unmarried women who report less recent sexual intercourse’ – otherwise the study would maintain the status quo biases against these women and would not substantively contribute new content to the collective research base. 

Note, we reordered the subgroups in the Methods section to focus on unmarried women with less recent sexual intercourse. 

2. If the aim of the study was to assess “trends in … contraceptive utilization and unmet need”, then, I would expect to see these trends presented in the abstract. The authors indicate that they used data collected in multiple surveys over the period 2014 to 2019; so, I expected to see some trend analyses presented, and I would be interested to know if there was a significant increasing or decreasing trend or whether there was no change over the years in contraceptive utilization and unmet need. This is not provided. Instead, the authors focus on reporting on 2019 indicators which makes it difficult to tell if the analysis was to assess trends or just contraceptive utilization in 2019. Also, the reporting on the trends in unmet need comes at the extreme end of the results sub-section, and presented in a more generic format.

>>AUTHORS: Thank you for the comment; however, based on the content of the original submission, the previous group of reviewers recommended that the term “Trends” be added to the title. See below. “Trend” was not part of the original title, because analysis of trend was not the primary objective of the study. Given your feedback and our intended focus on level and contextual factors, the Authors decided to remove “Trend” from the title and description of objectives. 

Original Title: Contraceptive Demand and Utilization by Unmarried, Sexually Active Women in Kenya: A Multilevel Regression Analysis

From January 3, 2022:

Reviewer #3: Review Outcome

• Suggest the title to be modified

• Suggested title: Trends in Contraceptive Demand and Utilization Among Sexually Active Unmarried Women in Kenya: A Multilevel Regression Analysis

3. I realize that the issue of emergency contraception is singled out in the abstract. Why was this singled out and not considered as one of the contraceptive methods, in the same way the other short-term methods were handled.

>>AUTHORS: Thank you for the feedback. As mentioned in the Abstract and further discussed in the Introduction and Methods, the PMA questionnaire contains a unique question - “Have you used emergency contraception at any time in the last 12 months?”. This is one of the only datasets in the world that includes this question. Emergency contraceptive are the primary post-intercourse contraceptive method; therefore, the typical “current contraceptive use” questions are less applicable and valid. Therefore, PMA added a more appropriate indicator for actual use by women. EC is singled out in the analysis, because PMA cleverly designed its questionnaire to provide a more valid estimate of genuine use by women. 

Results

1. Table 3 presents the contraceptive method mix by female Subgroup for the three surveys considered in the analysis (2014, 2017, 2019). Did the authors try to assess if the observed changes, as reported, depicted significant increases or decreases in the outcome over time? This question also applies to Figure 4 on unmet need.

>>AUTHORS: Thank you for the comment. The Authors did consider this analysis, but several overlapping issues led us to simply display the cumulative proportions and point estimates. First, contraceptive method mix is typically reported as cumulative proportions and several contraceptive methods had very small sample sizes; therefore, statistical inference would be limited. Secondly, the primary focus of the study was the level and contextual factors (see discussion above). The Methods Mix and Figures are unadjusted for these contextual factors. The Authors provided the Methods Mix table (and other descriptive indicators / figures) over time to provide background context to readers for better conceptual understand of the logistic models and experience of women in Kenya. [Note, if recommended, we can provide this statistical analysis, but it would require a lot of caveats and add to the length of the Methods section and overall study.]

---

## [Editor Report · Decision Letter 2]

12 Jun 2022

Contextual Factors associated with Contraceptive Utilization and Unmet Need Among Sexually Active Unmarried Women in Kenya: A Multilevel Regression Analysis

PONE-D-21-35162R2

Dear Dr. Nemser,

We’re pleased to inform you that your manuscript has been judged scientifically suitable for publication and will be formally accepted for publication once it meets all outstanding technical requirements.

Kind regards,

Joseph KB Matovu, Ph.D.

Academic Editor

PLOS ONE
---

## [Editor Report · Acceptance letter]

21 Jun 2022

PONE-D-21-35162R2 

Contextual Factors associated with Contraceptive Utilization and Unmet Need Among Sexually Active Unmarried Women in Kenya: A Multilevel Regression Analysis 

Dear Dr. Nemser:

I'm pleased to inform you that your manuscript has been deemed suitable for publication in PLOS ONE. Congratulations! Your manuscript is now with our production department. 

Kind regards, 

on behalf of

Dr. Joseph KB Matovu 

Academic Editor

PLOS ONE